# General features and contact modeling of *N*-body isolated resonances near threshold

Ludovic Pricoupenko

Sorbonne Université, CNRS-UMR 7600, Laboratoire de Physique Théorique de la Matière Condensée (LPTMC), F-75005 Paris, France.

## Abstract

*N*-body non efimovian bound or quasi-bound states for particles with short range interactions are considered in arbitrary dimensions. The different resonance regimes near the threshold are depicted by using a generalization of the effective range approximation. This two-parameter description can be used in various contexts from ultracold to hadronic physics. The universal character of these states makes it possible a formulation in terms of a contact model. The singularity at the contact imposes the introduction of a modified scalar product to solve the normalization catastrophe and to restore the self-adjoint character of the model. An equivalence with the standard scalar product used for realistic finite range models is derived.

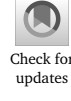

# 1 Introduction

## 1.1 General context

Near threshold $N$-body bound and quasi-bound states, denoted in short $N$-body resonances play an important role in scattering properties. They are observed and/or predicted for systems with short range interparticle potentials in hadronic and nuclear physics, condensed matter and in ultracold atomic physics [1–9]. In the context of the many-body problem, they induce correlations at large scales in many-body systems [10, 11]. Despite the considerable differences in energy scales and the peculiar short distance physics specific to these various

systems, they share universal properties. Interestingly, ultracold atoms experiments provide marvelous opportunities to explore in depth these highly correlated states with the possible tuning of the interactions via magnetic and optical Feshbach resonances [12]. For instance depending on the statistics, the number and the mass of the particles, Efimov states can be observed or predicted near the unitary limit of the $s$-wave two-body interaction [13–18].

At the heart of this universality is the scale invariance in a wide interval of lengths, a property that implies the separability in the hyperangle ($\Omega$) and hyperradius ($\rho$) coordinates for several orders of magnitude. In the separability region, the hyperradial problem can be mapped to a radial two dimensional (2D) Schrödinger equation with an inverse square potential proportional to $s^2/\rho^2$. The strength of this generalized kinetic barrier is obtained from the hyperangular eigenvalue problem by using the behavior of the wave function near the contact of two interacting particles. The Efimov effect, characterized by a discrete scale invariance of the spectrum at vanishing energy emerges when the hyperradial potential is attractive, i.e. when the index $s$ is imaginary. The beauty of this effect is that the long range attractive inverse square potential emerges from the short distance behavior of the wave function itself. Besides the Efimov effect which has attracted a lot of interest since its first observation in 2006 [15], isolated $N$-body resonances occur when the tail of this potential is repulsive in the separability region. Such resonances exist for sufficiently attractive interactions in the short hyperradius region where the wave function is no longer separable. Instead of using realistic finite range reference Hamiltonians, a way to obtain universal laws in such regime is to use a contact model where by construction, the short distance physics which is not universal, is absent from the formalism. First studies of this regime using a contact model were done in Refs. [19–21]. However, due to the normalization catastrophe, i.e. the fact that localized contact states are not square integrable when $s \geq 1$, the model was not believed to be tractable for arbitrary resonances and the universality of these states was questionable. Nevertheless, for a large class of resonances, one expects also near threshold universal laws as predicted in Ref. [22–25].

## 1.2 First advances obtained in Ref. [26]

Part of the answers to these pending issues has been given recently in Ref. [26] in the context of the unitary limit. In this last reference a contact model for isolated $N$-body resonances has been proposed for all real values of the index $s$ (in what follows $s$ will be always considered as positive) when part of the particles interact resonantly in the $s$-wave. The present paper permits one to replace the results of Ref. [26] in a more general context. In this introduction, it is thus necessary to recall the main results obtained in Ref. [26] as follows.

In a close analogy with the two-body scattering in high partial wave, it was shown that two parameters are in general needed to describe low energy states. The normalization catastrophe was solved by introducing a modified scalar product which restores also the self-adjoint character of the contact Hamiltonian. Considering the wronskian of two reference states (i.e. eigenstates of the finite range Hamiltonian associated with the contact model), this scalar product was shown to take into account the small hyperradius contribution of the reference states. A two parameter law for the binding energies was given for all values of $s$. Moreover, long lived quasi-bound states were predicted when $s \geq 1$, for vanishing detuning from the threshold. All the results obtained in Ref. [26] were qualitatively interpreted by what can be called the 3D mapping, meaning the formal equivalence of the hyperradial $N$-body problem for half-integer values of the index $s = \ell + 1/2$, with the radial problem of a two-body system in 3D with a resonant $\ell$ wave interaction.

## 1.3   The need of a more general contact model

The scope of application of the contact model of Ref. [26], concerns systems in the vicinity of the unitary regime where part of the pair of particles interact resonantly in the $s$-wave in a three-dimensional (3D) space. It appears that extension of the contact model to situations where there is no unitary limit or in a one-dimensional (1D) or a 2D space is possible. Another issue is how one can conciliate the two-parameter model of a $N$-body resonance in Ref. [26] with the single parameter models of Refs. [5, 19, 20] when $s < 1$ ? Always in the region $s < 1$, when $s = 1/2$, the 3D mapping for the contact model of Ref. [26] leads to the description of a broad $s$-wave resonance. There is no way to model the narrow resonant limit with this contact model: a generalization is thus expected. To end up this series of issues to solve, perhaps the most important one is to understand the link between the two parameters of the contact model and the reference model. More precisely, one can wonder how to conciliate an energy independent parameterization of the log-derivative of the wave function near the $N$-body contact (see Eq.(6) in Ref. [26]), with the effective range approximation where the energy dependence is explicit, as was done in this last reference.

## 1.4   Main results and outline of the paper

In Ref. [26] all the low energy properties were parameterized starting directly from a two-parameter contact model. The two parameters, i.e. the detuning from the resonance and the effective radius, were not directly linked to the behavior of the reference states. Here one adopts a more physical point of view in Sec. 2 by considering as a starting point, the properties of the reference states near the threshold. The reference model is introduced in this section together with the notion of separability region and by taking into account the possible occurrence of two-body resonances. The separability region is defined by an interval of the hyperradius $R_0 \leq \rho < R_{\text{sup}}$ and also hyperangles such that the particles do not feel the short range interactions. In the separability region the reference state is approximated by the product of a function depending on the hyperradius and of a function of the hyperangles of the $N$-body problem.

All the threshold properties are deduced in the subsequent sections from the log-derivative of the hyperradial function at the radius $R_0$ which defines the lower bound of the separability region. It is shown that the occupation of the small hyperradius region is related to the energy dependence of this log-derivative.

The hyperradial problem is characterized by an inverse square potential tail. The scattering problem in such a potential is revisited in Sec. 3 in the effective range approximation. The low energy properties are parameterized in terms of the generalized scattering length and of the range parameter. The different non efimovian regimes of resonance are then analyzed in Sec. 4.

Using the results of the preceding sections, the contact model is introduced in Sec. 5. It models $N$-body resonances in the vicinity of the threshold and when few-body interactions are of short range. For $s > 1$, one recovers the contact model of Ref. [26] but for $s < 1$, the model of Ref. [26] appears as a particular case where the range parameter is not large. In Sec.5.7, the modified scalar product introduced in Ref. [26] is generalized to encompass all the possible resonant regimes. A proof of its equivalence with the usual scalar product associated with the reference model is derived.

To have a qualitative picture of the spectrum when the upper bound of the separability region $R_{\text{sup}}$ is finite in absence of shallow dimer, a box model is introduced in Sec. 6. One obtains a branch similar to that of an Efimov spectrum.

In Sec. 7 a simple finite range 1D model is used to compare the standard normalization with that obtained by using the modified scalar product in the contact model.

## 2 Reference model

### 2.1 Separability region

#### 2.1.1 Generic case

One considers $N$ point-like particles that evolve in a space of dimension $D$. The mass of the particle $i$ is $m_i$ and its spatial coordinates are $\mathbf{r}_i$. The interparticles interactions are characterized by the radius $R_{\text{pot}}$. The Hamiltonian describing the system, denoted as the reference model, is supposed to have only one open channel. More precisely, deeply bound composite particles in the reference model are considered as stable and structureless. In other words, there is no possible break-up and rearrangement of these composite particles in few body collisions. In the generic case of a $N$-body resonance one considers a situation where there is no $M$-body resonance ($M < N$) in the system and if a $N$-body state exists it can be considered as brunnian (a generalization of the notion of Borromean states for the $N$-body problem, i.e. no subsystem is a bound state).

In this paper, the systems are considered in the center of mass frame. For convenience, one introduces the dimension of the configuration space of the particles which is given by

$$d = (N-1)D. \tag{1}$$

The positions of the particles are described by the $d$-dimensional hyperradius vector $\boldsymbol{\rho}$ equals to the set of Jacobi coordinates $\boldsymbol{\eta}_j$ ($j = 1 \dots N-1$), which are the relative coordinates between the center of mass $\mathbf{C}_j$ of the set of particles $1 \dots j$ of total mass $M_j$ and the particle $j+1$:

$$\boldsymbol{\eta}_j = \sqrt{\frac{\mu_j}{m_{\text{r}}}} \left( \mathbf{r}_{j+1} - \mathbf{C}_j \right). \tag{2}$$

In Eq. (2), $m_{\text{r}}$ is an arbitrary reference mass and $\mu_j$ is the reduced mass of the relative particle:

$$\mu_j = \frac{m_{j+1} M_j}{M_{j+1}}. \tag{3}$$

Other sets of Jacobi coordinates can be defined by permutations of the labels of the particles. In what follows, one will use the separability with respect to the hyperradius $\rho = \|\boldsymbol{\rho}\|$ and the hyperangles $\boldsymbol{\Omega}$ which parameterize the unit hypervector $\boldsymbol{\rho}/\rho$. The hyperradius can be expressed as a function of the positions $\{\mathbf{r}_i\}$ and the center of mass of the particles $\mathbf{C}_N$ with

$$\rho^2 = \sum_{i=1}^{N-1} \eta_i^2 = \sum_{i=1}^{N} \frac{m_i}{m_{\text{r}}} (\mathbf{r}_i - \mathbf{C}_N)^2. \tag{4}$$

In this coordinates system, the kinetic operator in the center of mass frame can be expressed in terms of the hyperradial kinetic operator

$$T_\rho = -\frac{\hbar^2}{2m_{\text{r}}} \left( \partial_\rho^2 + \frac{d-1}{\rho} \partial_\rho \right), \tag{5}$$

and of the Laplacian $\Delta_{\boldsymbol{\Omega}}$ acting on the hypersphere of radius unity:

$$H_0 = \frac{-\hbar^2}{2m_{\text{r}}} \Delta_\rho = T_\rho - \frac{\hbar^2}{2m_{\text{r}}} \frac{\Delta_{\boldsymbol{\Omega}}}{\rho^2}. \tag{6}$$

A standard approach in few-body physics is to expand the reference state in terms of the hyperspherical harmonics $\Phi^{[\lambda]}(\boldsymbol{\Omega})$ [27]:

$$\langle \boldsymbol{\rho} | \Psi_{\text{ref}} \rangle = \sum_{[\lambda]} \rho^{1-\frac{d}{2}} F^{[\lambda]}(\rho, E) \Phi^{[\lambda]}(\boldsymbol{\Omega}). \tag{7}$$

In Eq. (7) the notation $[\lambda]$ gathers all the quantum numbers that define the system. The hyperspherical harmonics are eigenstates of the Laplacian on the unit hypersphere $\Delta_{\boldsymbol{\Omega}}$:

$$-\Delta_{\boldsymbol{\Omega}}\Phi^{[\lambda]}(\boldsymbol{\Omega}) = \Lambda\Phi^{[\lambda]}(\boldsymbol{\Omega}). \tag{8}$$

Near the threshold of a resonance, for a large class of systems, one component in the sum of Eq. (7) dominates for a sufficiently large hyperradius and one can use a separable approximation for $\rho > R_0$ where the separability radius $R_0$ is of the order of the potential radius $R_{\text{pot}}$ [1]:

$$\langle \boldsymbol{\rho} | \Psi_{\text{ref}} \rangle \simeq \rho^{1-\frac{d}{2}} F(\rho, E)\Phi(\boldsymbol{\Omega}). \tag{9}$$

Without loss of generality, the hyperspherical function will be always normalized on the hypersphere: $\langle \Phi | \Phi \rangle = 1$. In the separable approximation Eq. (9), one excludes all the configurations where the hyperradius $\rho_M$ of $M < N$ particles is of the order of $R_{\text{pot}}$ or smaller.

The radial pre-factor in the right-hand side of Eq. (9) has been chosen to have an effective 2D radial equation in the separability region

$$-\frac{\hbar^2}{2m_{\text{r}}}\left(\partial_\rho^2 + \frac{\partial_\rho}{\rho} - \frac{s^2}{\rho^2}\right)F(\rho, E) = EF(\rho, E). \tag{10}$$

In Eq. (10), the index $s$ is defined by

$$s^2 = \Lambda + (d/2 - 1)^2, \tag{11}$$

where $\Lambda$ is the eigenvalue of the dominant component in Eq. (7) obtained from Eq. (8) and $s$ is chosen positive when it is real. In the generic case, the eigenvalues of Eq. (8) are positive and given by

$$\Lambda = K(K + d - 2), \tag{12}$$

where the integer $K \geq 0$ is the hypermoment. Equations (11) and (12) can be combined to give

$$s = \left| K + \frac{d}{2} - 1 \right|. \tag{13}$$

Negative values of $\Lambda$ can be obtained in presence of two-body $s$-wave resonant interactions in a 3D space.

Equation (10) coincides with the radial equation (outside the potential radius) of a single-particle in a 2D space with an effective angular momentum $\hbar s$. This formal equivalence, Eq. (9) can be qualified as a 2D mapping of the initial $N$-body problem. Equivalently, the 3D mapping is obtained with the change of function

$$F(\rho, E) = \sqrt{\rho} f(\rho, E), \tag{14}$$

leading formally to the radial equation (outside the potential radius) of a 3D two-body problem with the angular momentum $\hbar(s - \frac{1}{2})$.

The continuity of the log-derivative of the hyperradial reference function at $\rho = R_0$ permits one to replace the effect of the short range interactions by a condition at the border of the separable region. Quite generally, for a state of energy $E = \frac{\hbar^2 k^2}{2m_{\text{r}}}$, it can be written as

$$\left. \frac{\rho \partial_\rho F(\rho, E)}{F(\rho, E)} \right|_{\rho = R_0} = -\upsilon(k^2 R_0^2), \tag{15}$$

where the function $v(k^2 R_0^2)$ depends on the details of the reference model at short distance. It will be assumed in all the subsequent study that the energy of the system $E$ is small in absolute value with respect to the high energy scale of the reference model:

$$E_0 = \frac{\hbar^2}{2 m_{\rm r} R_0^2}. \tag{16}$$

Consequently, the right-hand side of Eq. (15) can be approximated in the limit $|k R_0| \ll 1$ by

$$v(k^2 R_0^2) \simeq v_0 + v_0' k^2 R_0^2 + \dots \tag{17}$$

### 2.1.2 Occurrence of resonant two-body interactions

The presence of a two-body resonant interaction changes radically the structure of the separability region and the eigenvalue problem on the hypersphere in Eq. (8). When $D = 3$, for a $s$-wave resonant interaction between two particles $(ij)$ of reduced mass $\mu$, with a scattering length $a_{\rm 3D}$, there is a boundary condition near the contact of the particles which is given by the Bethe-Peierls condition:

$$\Psi_{\rm ref}(\mathbf{r}_1 \dots \mathbf{r}_N) \simeq A \times \left( \frac{1}{r_{ij}} - \frac{1}{a_{\rm 3D}} \right). \tag{18}$$

In Eq. (18) $|a_{\rm 3D}| \gg R_{\rm pot}$, the relative radius $r_{ij}$ is small with respect to all the lengths in the system except the potential radius: $r_{ij} > R_{\rm pot}$ and $A$ depends on the other coordinates. For a simplified analysis, the two-body behavior in Eq. (18) is extended formally to arbitrarily small values of $r_{ij}$. In absence of $M$-body resonance in the system ($N > M \geq 3$) one recovers the separability at the unitary limit $|a_{\rm 3D}| = \infty$, but the eigenvalue problem on the hypersphere changes due to the singularity at the vicinity of the contact of two interacting particles. The eigenvalues $\Lambda$ in Eq. (8) are obtained in the zero-range approximation of the two-body interaction by imposing the contact condition $\Phi(\mathbf{\Omega}) \propto \frac{\rho}{r_{ij}}$ when $r_{ij} \to 0$ for each pair $(ij)$ interacting resonantly in the $s$-wave. Except for $N = 3$, there is no general solution such as Eq. (12) of this eigenvalue problem [20]. For a large but finite value of $|a_{\rm 3D}|$, the reference wave function behaves as in the unitary limit for intermediate values of the hyperradius $R_{\rm pot} < \rho \ll |a_{\rm 3D}|$. In this interval of the hyperradius and again in absence $M$-body resonance ($N > M \geq 3$), for an energy $E$ in the interval $\hbar^2/(2\mu a_{\rm 3D}^2) \ll |E| \ll E_0$ the reference wave function is separable with the same index $s$ as in the unitary limit. Moreover, when the reference state is brunnian (domain $a_{\rm 3D} < 0$) with an energy sufficiently near the threshold $|E| \ll \frac{\hbar^2}{2\mu a_{\rm 3D}^2}$, one recovers the separability for $\rho \gg |a_{\rm 3D}|$ with the index $s$ obtained this time from Eq. (12).

In a 2D space, a two-body scattering resonance in the $s$-wave for a pair of particles $(ij)$ leads to a logarithmic behavior in the limit of small interparticle radius $r_{ij}$ and Eq. (18) is replaced by

$$\Psi_{\rm ref}(\mathbf{r}_1 \dots \mathbf{r}_N) \simeq A \times \ln \left( \frac{r_{ij}}{a_{\rm 2D}} \right). \tag{19}$$

Near the resonance, the 2D scattering length $a_{\rm 2D}$ is large, but contrary to the 3D case, there is no scale invariance. Moreover there is always a shallow dimer in the resonant limit and thus no brunnian state is possible [28]. Consequently, in the resonant limit, there is no separable region. The separability can be recovered only in the non resonant case for an hyperradius $\rho \gg a_{\rm 2D}$ and for states of energy $|E| \ll \frac{\hbar^2}{2 m_{\rm r} a_{\rm 2D}^2}$: a situation which corresponds to a value of $a_{\rm 2D}$ of the order of $R_{\rm pot}$.

In a 1D space, the two-body scattering resonance in the even sector occurs for a vanishing 1D scattering length $a_{\rm 1D} \simeq 0$ and the behavior of the reference state in the limit of a small $r_{ij}$ is given by

$$\Psi_{\rm ref}(\mathbf{r}_1 \dots \mathbf{r}_N) \simeq A \times \left( r_{ij} - a_{\rm 1D} \right). \tag{20}$$

In the vicinity of the two-body resonance there is no shallow dimer. One recovers at the resonance, the scale invariance and the separability with again a modification of the eigenvalue problem on the hypersphere. An example of state without $N$-body resonance which exhibits this scale invariance is given by the $N$-boson problem in the Tonks regime [29].

Two-body resonant interactions in a higher partial wave $\ell \geq 1$ are characterized by a typical length $L$ linked to the range term which is of the order of, or larger than the potential radius [30]. Then, for a brunnian state of energy much smaller than $\hbar^2/(m_r L^2)$, the situation does not basically differs from the case of Sec. 2.1.1.

In all these situations one can identify two lengths $R_0$ and $R_{\text{sup}}$ such that there is an hyper-angle/hyperradius separability of the reference wave function as in Eq. (9) in a wide range of scales of the hyperradius

$$R_0 < \rho < R_{\text{sup}}, \tag{21}$$

and when also $r_{ij} > R_0$ for all interacting pairs $ij$.

## 2.2 Resonance condition

In analogy with the two-body $s$-wave resonant scattering in a 3D space, one adopts here the definition of a $N$-body resonance by the threshold at which a $N$-body bound state has a vanishing energy, i.e. $|kR_0| \to 0$ in Eq. (15). Assuming that the ratio $R_{\text{sup}}/R_0$ is sufficiently large, one can consider the limit $R_{\text{sup}} = \infty$. The bound state solution of Eq. (10) for $\rho > R_0$ is then well approximated by the Macdonald function in the separability region:

$$F(\rho, E) = \mathcal{A} K_s(q\rho). \tag{22}$$

In Eq. (22), $\mathcal{A}$ is a normalization constant and the binding wavenumber $q$ is given by $q = \sqrt{-2m_r E}/\hbar$. Using the fact that

$$\left. \frac{x \partial_x K_s(x)}{K_s(x)} \right|_{x=0} = -s, \tag{23}$$

one deduces from Eq. (15) that for $R_{\text{sup}} = \infty$, the zero energy $N$-body resonance i.e. the threshold of the resonance occurs when

$$v_0 = s. \tag{24}$$

In what follows the detuning from the resonance will be denoted by

$$\delta = s - v_0. \tag{25}$$

## 2.3 Occupation of the small hyperradius region

Using the fact that the reference Hamiltonian is self-adjoint, one can deduce the contribution to the normalization of the small hyperradius region in terms of the wronskian:

$$\int_{\rho < R_0} d\mu \, |\langle \boldsymbol{\rho} | \Psi_{\text{ref}}(E) \rangle|^2 = \lim_{E' \to E} \frac{\hbar^2 R_0^{d-1}}{2m_r} \frac{\int d\Omega \, W\left[\langle \boldsymbol{\rho} | \Psi_{\text{ref}}(E') \rangle^*, \langle \boldsymbol{\rho} | \Psi_{\text{ref}}(E) \rangle, \rho = R_0\right]}{E' - E}. \tag{26}$$

To obtain this last equation, one has used the fact that for realistic potentials, the reference wave function and its hyperradial derivative vanish at the origin $\rho = 0$. In Eq. (26), $d\mu = \rho^{d-1} d\rho \, d\Omega$ where $d\Omega$ is the measure on the hypersphere and the notation $W[f, g, \rho = R_0] = f \partial_\rho g - g \partial_\rho f$ is the wronskian of the functions $f$ and $g$ with respect to the variable $\rho$, considered here at $\rho = R_0$.

Using the separability approximation of the reference state in Eq. (9) at $\rho = R_0$ and Eq. (15), one obtains from Eq. (26) the contribution to the norm, of the small hyperradius region in the low energy limit ($|E| \ll E_0$):

$$\int_{\rho < R_0} d\mu \, |\langle \boldsymbol{\rho} | \Psi_{\text{ref}}(E) \rangle|^2 \simeq v_0' |R_0 F(R_0, E)|^2 \,. \tag{27}$$

From Eq. (27) one obtains the inequality:

$$v_0' > 0 \,. \tag{28}$$

Consider now a bound state of energy $E$ when $R_{\text{sup}} = \infty$. The probability to find the particles in the region $\rho < R_0$ is

$$\mathcal{P}_{<R_0}(E) = \frac{\int_{\rho < R_0} d\mu \, |\langle \boldsymbol{\rho} | \Psi_{\text{ref}}(E) \rangle|^2}{|\langle \Psi_{\text{ref}}(E) | \Psi_{\text{ref}}(E) \rangle|^2} \,, \tag{29}$$

and at the threshold of the resonance, using Eq. (A.7), one finds:

$$\lim_{E \to 0} \mathcal{P}_{<R_0}(E) = \begin{cases} \frac{1}{1 + \frac{1}{2(s-1)v_0'}} \,, & \text{if} \quad s > 1 \,, \\ 0 \,, & \text{otherwise.} \end{cases} \tag{30}$$

Hence, there is no abrupt change in the occupation of the small hyperradius region at the critical value $s = 1$. Instead, for a fixed value of $v_0'$ and for increasing values of the index $s$, one has a continuous increase of the occupation in this region.

## 3 Scattering process in an inverse square potential

### 3.1 Partial wave amplitude

The wave function in Eq. (9) can be interpreted as the eigenstate of a single particle in a symmetric central potential in a $d$-dimensional space. In this point of view, one can consider the scattering problem of the particle on a central potential with a tail having an inverse square law.

At negative energy $E = -\frac{\hbar^2 q^2}{2m_r} < 0$ and in the separability region, the hyperradial function is a linear combination of the two modified Bessel functions

$$F(\rho, E) = \mathcal{A}(E) K_s(q\rho) + \mathcal{B}(E) I_s(q\rho) \,. \tag{31}$$

When $R_{\text{sup}} = \infty$, a bound state of binding energy $E_B$ is such that $\lim_{\rho \to \infty} F(\rho, E_B) = 0$ and thus $\mathcal{B}(E_B) = 0$. The hyperradial scattering function at energy $E = \frac{\hbar^2 k^2}{2m_r} > 0$ of this single-particle system is obtained by performing the analytical continuation of Eq. (31) with $q = -ik$ for $\rho > R_0$. The ratio $\mathcal{A}/\mathcal{B}$ can then be redefined in terms of a partial wave amplitude (the other part of the scattering amplitude is a function of the hyperangles) at energy $E$ with:

$$f_s(E) = \frac{\pi \mathcal{A}(E) e^{i\pi s} k^{2-d}}{2\mathcal{B}(E)} \,. \tag{32}$$

Then for $E > 0$:

$$F(\rho, E) = i^{-s} \mathcal{B}(E) \left[ J_s(k\rho) + i k^{d-2} f_s(E) H_s^{(1)}(k\rho) \right] \,, \tag{33}$$

the factor $k^{d-2}$ has been inserted for convenience by considering the expansion of the $d$-dimensional plane wave on the hyperspherical harmonics [27, 31] with a first term corresponding to the lowest hypermoment $K = 0$, proportional to $(k\rho)^{1-d/2} J_{\frac{d}{2}-1}(k\rho)$, and also the

expression of the $d$-dimensional Green's function of the free Schrödinger equation at energy $E$ for a source term $-\delta(\boldsymbol{\rho})$, i.e.

$$G_d(\rho, E) = \frac{-i\pi m_r}{(2\pi)^{d/2}\hbar^2} \left(\frac{k}{\rho}\right)^{\frac{d}{2}-1} H^{(1)}_{\frac{d}{2}-1}(k\rho).$$ 
(34)

Coming back to the $N$-body problem, one has to be aware that Eq. (33) does not take into account the possible $M$-body scattering process ($M < N$) but corresponds somehow to a pure $N$-body scattering.

## 3.2 Scattering phase shift

The link between the partial wave amplitude and the scattering phase shift is obtained by considering another possible expression of the hyperradial function for $\rho > R_0$ in Eq.(33) by using Eq. (A.2):

$$F(\rho, E) = \frac{i^{-s}\mathcal{B}(E)}{2}\left[\left[1 + 2if_s(E)k^{d-2}\right]H^{(1)}_s(k\rho) + H^{(2)}_s(k\rho)\right].$$ 
(35)

In absence of interaction, the partial wave amplitude is zero, so that at low energy ($kR_0 \ll 1$) and large distance ($k\rho \gg 1$), the hyperradial function is

$$F_{\text{free}}(\rho, E) \simeq \frac{i^{-s}\mathcal{B}(E)}{\sqrt{2\pi k\rho}}\left(e^{i(k\rho - \frac{\pi}{4} - \frac{\pi s}{2})} + e^{-i(k\rho - \frac{\pi}{4} - \frac{\pi s}{2})}\right),$$ 
(36)

where one has used Eq. (A.5) and Eq. (A.4). The interaction introduces the scattering phase-shift $\delta_s(E)$ such that in the low energy and large distance limit:

$$F(\rho, E) \simeq \frac{i^{-s}e^{i\delta_s(E)}\mathcal{B}(E)}{\sqrt{2\pi k\rho}}\left(e^{i(k\rho - \frac{\pi}{4} - \frac{\pi s}{2} + \delta_s(E))} + e^{-i(k\rho - \frac{\pi}{4} - \frac{\pi s}{2} + \delta_s(E))}\right).$$ 
(37)

The comparison of Eq. (37) with Eq. (35) permits one to define the scattering phase shift $\delta_s(E)$ in terms of the partial wave amplitude

$$e^{2i\delta_s(E)} = 1 + 2if_s(E)k^{d-2},$$ 
(38)

so that

$$f_s(E) = \frac{k^{2-d}}{\cot\delta_s(E) - i} = -ik^{2-d}e^{i\delta_s(E)}\sin\delta_s(E).$$ 
(39)

The form of the function $\cot\delta(E)$ in the low energy limit is deduced from the log-derivative of the reference hyperradial function at $\rho = R_0$ given in Eq. (15). Using Eq. (A.3) and Eq. (A.4), one finds

$$\cot\delta_s(E) = \cot\pi s - k^{-2s}u_s(k^2),$$ 
(40)

where the function

$$u_s(k^2) = \frac{k^{2s}}{\sin\pi s} \times \left.\frac{v(x^2)J_{-s}(x) + xJ'_{-s}(x)}{v(x^2)J_s(x) + xJ'_s(x)}\right|_{x=kR_0},$$ 
(41)

characterizes the effect of the interactions and is regular at $k = 0$ for non integer values of $s$.

From Eq. (41), and the relations in Eqs. (A.3,A.1), one deduces as expected that the poles of the partial wave amplitude in Eq. (39) correspond to the binding energies which satisfy also

$$\left.\frac{x\partial_x K_s(x)}{K_s(x)}\right|_{x=qR_0} = -v(-q^2R_0^2).$$ 
(42)

### 3.3 Scattering parameters in the low energy limit

In the low energy limit, the function $u_s(k^2)$ can be expanded in a series of $k^2$. Limiting the low energy expansion of $u_s$ at the order of $k^2$, the partial wave amplitude permits one to define the scattering parameters at low energy:

$$u_s(k^2) = \frac{1}{\xi_s} + \alpha_s k^2 + C(s, k^2). \tag{43}$$

In this last equation, the term $C(s, k^2) = O(k^4)$ is the remainder of the expansion of the function $u_s(k^2)$ and the two first terms in the right-hand-side, define the effective range approximation. In what follows, the parameter $\xi_s$ will be denoted as the generalized scattering length and $\alpha_s$, as the range parameter. From Eq. (41), one has

$$\xi_s = \frac{\pi R_0^{2s}}{s 4^s \Gamma(s)^2} \left( \frac{v_0 + s}{v_0 - s} \right), \tag{44}$$

$$\alpha_s = \frac{s 4^{s-1} \Gamma(s)^2}{\pi} R_0^{2-2s} \left( \frac{s - v_0}{s + v_0} \right) \left[ \frac{4v_0' + \frac{2 + v_0 - s}{s - 1}}{s - v_0} - (s \to -s) \right]. \tag{45}$$

The generalized scattering length gives the coefficient of the Wigner's threshold law for the partial wave amplitude:

$$f_s(E) \underset{k \to 0}{\simeq} -k^{2s+2-d} \xi_s. \tag{46}$$

As expected, using Eq. (24), one finds that the generalized scattering length is arbitrarily large at resonance $|\xi_s| = \infty$ whereas the range parameter is

$$\alpha_s^{\text{res}} = \frac{4^{s-1} R_0^{2-2s} \Gamma(s)^2}{\pi(s-1)} \left( 1 + 2(s-1)v_0' \right). \tag{47}$$

In the relevant limit of a small detuning $|\delta| \ll 1$, the two scattering parameters in Eq. (44,45) can be written:

$$\xi_s = -\frac{\pi R_0^{2s}}{s 4^s \Gamma(s)^2} \left( \frac{2s - \delta}{\delta} \right), \qquad \alpha_s \simeq \frac{\alpha_s^{\text{res}}}{\left( 1 - \frac{\delta}{2s} \right)^2}. \tag{48}$$

For $s > 1$, the range parameter is always positive and from Eq. (28) one obtains the generalization of the width radius inequality already obtained for high partial wave scattering [30]:

$$\alpha_s^{\text{res}} R_0^{2(s-1)} > \frac{\Gamma(s)^2}{4^{(1-s)}(s-1)\pi}. \tag{49}$$

This bound and also Eq. (30) generalize the results found in Ref. [32] for the two-body problem in an arbitrary dimension, but here with a continuous value of $s$.[1]

### 3.4 Mapping to the two-body problem

Let us consider a two-body scattering process in the center of mass frame in a 2D space. For an incident plane of wave vector $\mathbf{k}$, the scattering state $|\Psi_{2D}\rangle$ can be written asymptotically for $kr \gg 1$ [33]:

$$\Psi_{2D}(\mathbf{r}) \simeq e^{i\mathbf{k} \cdot \mathbf{r}} + \sqrt{\frac{i}{kr}} f_{2D}(k, \theta) e^{ikr}, \tag{50}$$

---

[1]See Eq. (12) and Eq. (14) of Ref. [32] with the equivalences: $s \equiv d/2 + l - 1$, $\xi_s \equiv a_{L,d}$ and $\alpha_s \equiv -r_{L,d}/2$.

where $\mathbf{r}$ denote the relative coordinates, $\theta = \angle(k, \mathbf{r})$ and the partial wave amplitudes $f_{2D}^{[n]}(E)$ for the angular momentum $n\hbar$ can be defined by

$$f_{2D}(k, \theta) = \sqrt{\frac{2}{\pi}} \sum_{n=-\infty}^{\infty} f_{2D}^{[n]}(E) e^{in\theta} . \tag{51}$$

Using this last definition, the scattering state can be expressed for $r$ larger than the 2D potential radius, as:

$$\Psi_{2D}(\mathbf{r}) = \sum_{n=-\infty}^{\infty} i^n e^{in\theta} \left[ J_n(kr) + i f_{2D}^{[n]}(E) H_n^{(1)}(kr) \right] . \tag{52}$$

For an isotropic interaction, the 2D partial wave amplitudes satisfy $f_{2D}^{[-n]}(E) = f_{2D}^{[n]}(E)$ and can be related to the partial wave amplitude in Eq. (39) with

$$f_{2D}^{[n]}(E) = f_n(E) k^{d-2} , \tag{53}$$

which is a consequence of the 2D mapping.

In the 3D two-body problem with an isotropic interaction potential, the partial wave amplitude is defined from the scattering phase shift by

$$f_{3D}^{[\ell]}(E) = \frac{1}{k \cot \delta(E) - ik} . \tag{54}$$

The partial wave amplitude is thus also proportional to the 3D two-body partial wave amplitude in the $l$-wave when $s = \ell + \frac{1}{2}$ is an half-integer values with:

$$f_{3D}^{[\ell]}(E) = f_{\ell+\frac{1}{2}}(E) k^{d-3} . \tag{55}$$

## 4 Different regimes of resonances

The study of the shallow bound state near the resonance threshold permits one to identify different types of resonant regimes. Using the effective range approximation for $v(-q^2 R_0^2)$ in Eq. (43) gives an accurate approximation for the determination of the bound or quasi-bound state energies near threshold with the equation:

$$\frac{R_0^{2s}}{\xi_s} + \alpha_s R_0^{2(s-1)} \frac{E}{E_0} + C\left(s, \frac{E}{E_0}\right) = \frac{\left(-\frac{E}{E_0}\right)^s}{\sin \pi s} . \tag{56}$$

Complex solutions of Eq. (56) with a small negative imaginary part and a positive real part correspond to quasi-bound states. Equation (56) can be also written in terms of the binding wavenumber as

$$\frac{1}{\xi_s} - \alpha_s q^2 + R_0^{-2s} C(s, -q^2 R_0^2) = \frac{q^{2s}}{\sin \pi s} . \tag{57}$$

Let us make some comments about the singular behavior of the right-hand side of Eqs. (56,57) near integer values of $s$. In the limit $s = 0$, the singularity is compensated by the one of left-hand side where $\xi_s \sim \pi s$. In the limit $s = 1$, the singularity is this time compensated by the range term in the left-hand side where $\alpha_s \sim 1/(\pi(s-1))$. For higher integer values of $s$, the singularity is compensated by the remainder $C(s, -q^2)$ which plays the role of a counter term.

For a small detuning, the remainder can be always neglected for $s < 1 + \eta$ where $\eta > 0$ and is arbitrarily small in the zero energy limit. However, as one considers finite values of the detuning, it is chosen at an intermediary value between 0 and 1 typically $\eta \sim 0.2$. The exact expression of $C(s,z)$ depends on the $i-$th derivatives $v_0^{(i)}$ $i = 0 \ldots n$ which have been neglected for $i \geq 2$ in the effective range approximation. Then in this respect, it can be chosen arbitrarily with the sole aim of curing the singularity. For example a possible counterterm is $C(s,z) = 0$ for $s < 1 + \eta$ and for $s \geq 1 + \eta$:

$$C(s,z) = \frac{z^n}{\pi(s-n)}, \quad \text{where} \quad \begin{cases} n = \lceil s \rceil, & \text{if} \quad s > \lfloor s \rfloor + \eta, \\ n = \lfloor s \rfloor, & \text{otherwise.} \end{cases} \tag{58}$$

When $s = n \geq 2$ is an integer, Eq. (56) becomes

$$\frac{R_0^{2n}}{\xi_s} + \alpha_s R_0^{2n-2} \frac{E}{E_0} = \frac{1}{\pi} \left( \frac{E}{E_0} \right)^n \ln \left( -\frac{E}{E_0} \right). \tag{59}$$

For a given value of the index $s$ and of the separability radius $R_0$, the spectrum is defined from the values of $v_0'$ and of the detuning $\delta$.

## 4.1 Regimes of vanishing detuning

If the detuning $\delta$ is sufficiently small, the generalized scattering length $\xi_s$ is large. Due to the vicinity with the threshold, this corresponds to the situation where one finds a low energy bound (when $\xi_s > 0$) or possibly quasi-bound (when $\xi_s < 0$) state. In this small detuning limit, using Eq. (48), the range parameter can be approximated by its value at resonance only when $|\delta| \ll s$: a condition which is not satisfied in the vicinity of the Efimov threshold where $s \to 0$.

### 4.1.1 One-parameter resonant regime

When $s < 1$ the range term can be negligible in Eqs. (56,57) in the small energy limit $qR_0 \ll 1$. This happens in two situations: $i$) for a vanishing value of the detuning when the index is not too close to unity [$qR_0 \ll (qR_0)^s$ in Eqs. (56,57)], and/or $ii$) when

$$v_0' = \frac{1}{2(1-s)}. \tag{60}$$

In this regime, for a positive generalized scattering length (i.e. $\delta < 0$), there is one bound state with a one-parameter law for the binding wave number

$$q \simeq \left( \frac{\sin(\pi s)}{\xi_s} \right)^{\frac{1}{2s}}. \tag{61}$$

The binding energy $E_{1p}$ is then

$$E_{1p} = -4E_0 \left( \frac{s \sin(\pi s) \Gamma(s)^2 \delta}{\pi(\delta - 2s)} \right)^{1/s}. \tag{62}$$

This regime has been first defined and studied in Refs. [5, 19, 20]. Importantly, there is no quasi-bound state in this regime when $\delta > 0$.

### 4.1.2 Two-parameter resonant regime

The range parameter plays a increasing role for increasing values of the index $s$. For $s \geq 1$ it can never be neglected for vanishing values of the detuning and in the low energy limit $[qR_0 \gg (qR_0)^s$ in Eq. (56)]. Starting from the one-parameter law, valid for small values of the index $s$, the spectrum reaches a two-parameter law as $s$ increases. In the limit of a large index $s$, the binding wavenumber is given by:

$$q^2 \sim \frac{1}{\xi_s \alpha_s} . \tag{63}$$

Equation (63) is a very good approximation even when $s$ is larger but of the order of the unity. The binding energy is thus

$$E = E_r \simeq \frac{E_0 \delta}{v'_0 + \frac{1}{2(s-1)}} . \tag{64}$$

Still for $s > 1$, aside the shallow bound state solution which exists for a small and negative detuning $\delta$, i.e. for a large and positive generalized scattering length, when $\delta$ is small and positive, there is a long-lived quasi-bound state associated with the complex root of Eq. (56):

$$E = E_r - i \frac{\Gamma}{2} . \tag{65}$$

The resonance energy $E_r > 0$ can be approximated by Eq. (64) and the width $\Gamma$ by

$$\frac{\Gamma}{E_r} \simeq \frac{2\pi (s-1) 4^{1-s}}{(1 + 2(s-1) v'_0) \Gamma(s)^2} \left( \frac{E_r}{E_0} \right)^{s-1} . \tag{66}$$

This form of asymptotic behavior in $E_r^{s-1}$ was already found in Ref. [25] for half integer values of the index and more recently in Ref. [22] by using the Effective Field Theory formalism. In this last reference, the index $s$ is evaluated through the minimal energy $\hbar\omega\Delta$ of the system when considered in an isotropic harmonic trap of frequency $\omega$. The equivalence $\Delta = s + \frac{5}{2}$ applies both in the generic case [34][2] and in the unitary limit (see for instance Eq. (34) in Ref. [19]). In Ref. [22], an estimates was given for the life-time of near theshold $^3$He droplets of positive energy (i.e. $N \lesssim 29$). In this case, the scattering length is rather small and negative ($-13a_0$ in atomic units [35]), so that the brunnian resonance occurs in the generic case. The present derivation gives an overall suppression factor $4^{-s}/\Gamma(s)^2$ in the ratio $\Gamma/E_r$ not present in Ref. [22], where the reasoning was partly based on scaling properties. This enhances even more the life-time of the quasi-bound state when $s$ is large. Interestingly, the present derivation point out the crucial condition $E < E_0$ set by the separability radius $R_0$. If this condition is not met, even if the formula for $\Gamma/E_r$ in Eq. (66) gives a very small value due to the prefactor, this law is no longer valid. In Ref. [22], $E_0$ is estimated at the value 40 K which is much larger than the energy of the resonance $E_r = 0.0194 \times N = 0.56$ K at $N = 29$ in Ref. [36]. One can also use the mean radius of the droplet in Ref. [37] for an estimate of the order of magnitude of $R_0$ with $R_0 \sim 7.8$ Å and find $E_0 \sim E/2$. This shows that for a precise evaluation of $E_0$ one needs more informations about the many-body wave function than what is published in Refs. [36,37] and a more refined study is required to known wether or not the universal law is relevant in this case.

---

[2]One has the following correspondance with the notations of Ref. [34]: $N \equiv A$ (with $N = A - 1$ in [34]), $s \equiv \nu = \mathcal{L} + 1/2$ (see Eq. (2.16) in [34]) where $\mathcal{L} = L + (3N - 3)/2$ (see Eq. (2.14) in [34]) and $\Delta \equiv L + 3A/2$ (see Eq. (3.20) in [34]).

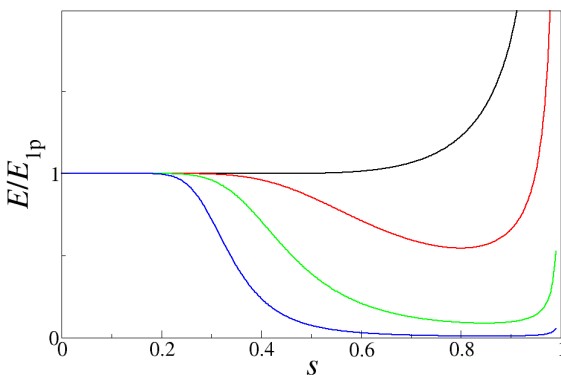

Figure 1: Ratio of the binding energy to the asymptotic binding energy $E_{1p}$ of Eq. (62), obtained for small values of $s$ and/or small detuning when the range term can be neglected. The detuning has been set to $s - v_0 = -10^{-2}$. Each line corresponds to a value of $v_0'$ (black: $10^{-3}$, red: 10, green: 100, blue: 1000).

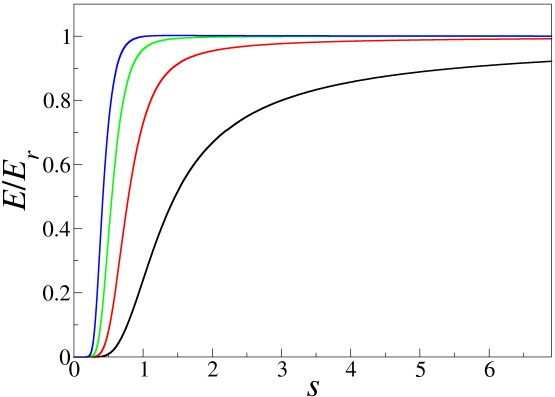

Figure 2: Ratio of the binding energy to the asymptotic energy $E_r$ in Eq. (68). The detuning has been set to $s - v_0 = -10^{-2}$. Each line corresponds to a value of $v_0'$ (black: $10^{-3}$, red: 10, green: 100, blue: 1000).

## 4.2 Regime of large range parameter

The limit of large range parameter values is especially interesting to study. For a fixed value of the index $s$, it is obtained in the limit $v_0' \gg 1$. In this regime, depending on the value of the detuning, the range term can be dominant with respect to the right-hand side of Eq. (57) even in the region $s < 1$. This regime encompasses the narrow resonance limit in the $s$-wave scattering for the 3D two-body problem, which occurs when the effective range is large and negative.

When $v_0' \gg 1$, the range parameter in Eq. (45) can be approximated by

$$\alpha_s \simeq \frac{4^s R_0^{2-2s} v_0' \Gamma(s+1)^2}{2\pi(s-\delta/2)^2} \, . \tag{67}$$

Using this last expression and neglecting the right-hand side of Eq. (56), one finds for a negative detuning $\delta < 0$ a bound state with the binding energy

$$\frac{E_r}{E_0} \simeq \frac{(2s-\delta)\delta}{2s v_0'} \, . \tag{68}$$

The range term is much larger than the right-hand side of Eq. (56) only for intermediate values of the detuning and a sufficiently large value of the index with the conditions:

$$(v_0')^{1-2s} \ll |\delta|^{2-2s}, \quad |\delta| \ll 1 \quad \Rightarrow \quad \frac{1}{2} < s. \tag{69}$$

Importantly, for a positive detuning $\delta > 0$, in the same limit there is a quasi-bound state

$$E = E_r - i\frac{\Gamma}{2}, \tag{70}$$

where the real part $E_r$ is given by Eq. (68) and the ratio between the width and the resonance energy is

$$\frac{\Gamma}{E_r} \simeq \frac{4^{1-s}\pi\left(1 - \frac{\delta}{2s}\right)^{s+1}}{\Gamma(s)^2 \delta^{1-s} v_0'{}^s}. \tag{71}$$

This ratio always vanishes near the threshold for $\delta \to 0^+$ when $s > 1$, where one recovers Eq. (66). More interestingly, it can be small even when $s < 1$, for a small but finite value of the detuning and in the limit $v_0' \gg 1$. Hence, long-lived quasi-bound states can be described by using the effective range approximation for $s < 1$. Nevertheless from Eq. (69), when $s < 1/2$ it is not possible to neglect the right-hand side of Eqs. (56,57) and Eqs. (68,71) are no longer valid. Moreover, the value of the detuning must be also sufficiently small. For instance, Eq. (68) leads to a spurious vanishing binding energy for $\delta = 2s^+$, a result which is not compatible with the decrease of a Macdonald function in Eq. (15). To conclude, Eqs. (68,71) are no longer valid in the vicinity of the Efimov threshold $s = 0$ where the range term is always negligible leading to the one-parameter law in Eq. (62). This excludes the possible occurrence of a quasi-bound state for small values of $s$. For $s < 1/2$ the range term is not dominant with respect to the right-hand side of Eq. (56). However, the disappearance of quasi-bound states is not abrupt at $s = 1/2$. One can notice that a thorough analysis of quasi-bound states in the effective range approximation for a 3D two-body $s$-wave resonance corresponding to the particular case where $s = 1/2$ has been done in Ref. [38].

The results of this section are illustrated in Figs. (1,2) where the spectrum is plotted as function of the index $s$ for different values of the slope $v_0'$ and a fixed value of the detuning. For increasing values of $v_0'$, the deviation of the spectrum from the one-parameter law of Eq. (62) occurs for a decreasing values of $s$ as shown in Fig. (1). *A contrario*, in the limit of large values of $v_0'$, the spectrum reaches the two-parameter law of Eq. (68) for decreasing values of $s$ with a crossover that occurs asymptotically for an index $s$ of the order of $1/2$, as expected.

## 4.3 Noticeable values of the index $s$

### 4.3.1 $s = 0$: A reminder of the 2D $s$-wave interaction

From Eq. (62) for $s = 0$, one obtains the binding energy $E = E^{(0)}$ with:

$$E^{(0)} = -4E_0 \exp\left(2/\delta - 2\gamma\right), \tag{72}$$

where $\gamma$ is the Euler's constant. One recovers the energy of a 2D dimer for a $s$-wave resonant interaction which can be deduced from Eq. (19) giving the binding wavenumber

$$q_{2D} = \frac{2e^{-\gamma}}{a_{2D}}. \tag{73}$$

Comparing this last equation with Eq. (72), one has the formal mapping $a_{2D} = R_0 \exp(-1/\delta)$. One can notice that this 2D physics can be achieved for two atoms of reduced mass $\mu$ in an harmonic atomic wave guide and interacting resonantly in the $s$-wave with the scattering length $a_{3D}$. The wave guide is characterized by a frequency $\omega_\perp$ and a transverse length $a_\perp = \sqrt{\hbar/\mu\omega_\perp}$. In this external potential, the 2D scattering length is given by [39–41]

$$a_{2D} \simeq a_\perp \exp\left(\frac{\mathcal{C}-\gamma}{2} - \frac{\sqrt{\pi}a_\perp}{2a_{3D}}\right), \qquad \mathcal{C} = 1.3605\dots \tag{74}$$

Concerning the range correction in the limit of vanishing values of the index $s$ such that $s \ll |\delta| \ll 1$, the parameter $\alpha_s$ cannot be approximated by Eq. (47). Instead, from Eq. (45) one finds:

$$\alpha_0 \simeq \frac{2v'_0 - 1}{\pi\delta^2}, \tag{75}$$

and the range correction for the binding energy is small:

$$E \simeq E^{(0)}\left(1 + \frac{(2v'_0 - 1)(E^{(0)})^2}{E_0^2\delta^2}\right). \tag{76}$$

### 4.3.2 $s = 1/2$: Back to the $s$-wave resonance

The case $s = 1/2$ is equivalent to the $s$-wave two-body resonant problem. Near the threshold the range parameter,

$$\alpha_{\frac{1}{2}} = (v'_0 - 1)R_0, \tag{77}$$

is related to the usual effective range $r_e$ with $r_e = -\frac{\alpha_{1/2}}{2}$ whereas the 3D scattering length $a$ is equal to $\xi_s$. The binding wavenumber is thus the solution of the usual equation of the effective range approximation

$$\frac{r_e q^2}{2} - q + \frac{1}{a} = 0. \tag{78}$$

The limit where $v'_0 \gg 1$ corresponds to a large and negative effective range which defines a narrow two-body $s$-wave resonance.

### 4.3.3 Critical value of the index $s = 1$

As already shown in Ref. [26] the index $s = 1$ corresponds to a critical value. In this limit, the range term is singular and annihilates the singularity in the right-hand side of Eq. (57), whereas the remainder $C(s, z)$ can be neglected. The eigenvalue equation Eq. (57) can be written

$$q^2 R_0^2 \ln\left(\frac{qR_0 e^{(\gamma - v'_0)}}{2}\right) = \delta. \tag{79}$$

For a large and positive generalized scattering length, i.e. a small and negative detuning $\delta < 0$, the binding energy is:

$$E \simeq \frac{-2E_0\delta}{W_{-1}\left(\frac{e^{2(\gamma - v'_0)\delta}}{2}\right)}, \tag{80}$$

where $W_{-1}$ is the Lambert function. For a positive and vanishing detuning $\delta > 0$, one obtains a long-lived quasi-bound state with the resonance position given by[3]

$$E_r \simeq \frac{-2E_0\delta}{W_{-1}\left(-\frac{e^{2(\gamma-v_0')}\delta}{2}\right)}, \tag{81}$$

and the ratio between the width and the resonance energy

$$\frac{\Gamma}{E_r} \simeq \frac{-2\pi}{W_{-1}\left(-\frac{e^{2(\gamma-v_0')}\delta}{2}\right)}. \tag{82}$$

As expected, in the limit of a large range parameter with $v_0' \gg -\ln(|\delta|)/2$, one recovers the asymptotic results of Eqs. (68,71).

The form of Eq. (79) has been found first in the context of hadronic physics in Ref. [24] by using Alt-Grassberger-Sandhas equations. In this last reference, the system is made of three particles with a single $s$-wave resonant pair and an example of quasi-bound state is given by the three particles $D\bar{D}\pi$. This situation corresponds indeed to the index $s = 1$ (see for instance section 6.1.3 in Ref. [18]). The same form as Eq. (79) was also given in Ref. [22]. The interest of the present derivation is to be more precise with respect to the number of free parameters in the universal laws of Eqs (80-82). Here, one finds two-parameter laws, with the two relevant independent parameters $\delta$ and $v_0'$. Equation (80) can be also relevant for the two neutrons halos studied in Refs. [8, 9]. The two neutrons are loosely bound and the halos are characterized by a two-neutron separation energy $S_{2n}$ (i.e. the energy needed to extract the two neutrons from the halo) much smaller than the binding energy of the core. In the case where the neutron-core scattering length (denoted by $a_c$) is also small with respect to the neutron-neutron scattering length $a_{nn} = -18.7$ fm [42], one recovers again a three-body system with a single $s$-wave resonant pair and the index $s = 1$ in the hyperradial function for $|a_{nn}| \gtrsim \rho \gtrsim |a_c|$. $^{22}$C is a good candidate of such two-neutron halo if one considers the upper bound $|a_c| < 2.8$ fm for the neutron-core scattering length (see section 2.4 of Ref. [4]). $^6$He ($S_{2n} = 975$ kev) is another example where the core is here the $\alpha$-particle and $a_c \sim 2.47$ fm, as measured in Ref. [43]. In this case, the $p$-wave resonant character of the interaction between the $\alpha$-particle and a neutron plays a crucial role in the binding of $^6$He (see Ref. [44] together with sections 3.7-3.10 in Ref. [4]). To conclude this paragraph, one has to notice that the law in Eq. (79) does not take into account the finite value of the neutron-neutron scattering length. The study of these particular halos and the link between the three-body parameters and the nuclear interaction is beyond the purpose of this work.

### 4.3.4 Case $s = 2$: A relevant limit for the three-body problem

The case $s = 2$ occurs frequently in few-body physics. It corresponds for instance to a resonant state made of three distinguishable particles without any resonant pairwise interaction [the value $s = 2$ is obtained for $\Lambda = 0, N = 3, D = 3$ in Eq. (11)]. In this case, as for larger integer values, only the imaginary part of the logarithmic term in Eq. (59) plays a role at the order of the effective range approximation. Then, one finds again Eq. (64) and Eq. (66) for the quasi-bound state. Similar laws were also derived in Ref. [25].

---

[3]In the limit $x \to 0^-$, the Lambert function can be approximated by $W_{-1}(x) \simeq \ln(-x) - \ln(-\ln(-x)) + \ldots$ Notice that there is an error in the expansion of the Lambert function used in Eq. (15) of Ref. [26] with an extra factor $e$ in the logarithm.

# 5 Contact model

## 5.1 Construction of the contact model

In the contact model, the separability is extended in the region of small hyperradius i.e. for $0 < \rho < R_{\text{sup}}$ where the hyperradial functions satisfy Eq. (10).

A contact model defines a set of eigenstates $\{|\Psi\rangle\}$ (contact eigenstates) associated with the set of the eigenstates of the reference model $\{|\Psi_{\text{ref}}\rangle\}$ (reference eigenstates). The contact eigenstates are solutions of the free Schrödinger equation everywhere except at the contact of two or more interacting particles and (almost) coincide with their corresponding reference states for $\rho > R_0$. This last property is obtained by imposing appropriate asymptotic condition(s) on the contact states at vanishing inter-particle distances.

In the contact model, the stationary Schrödinger equation for a state $|\Psi\rangle$ of energy $E$ is:

$$\left(T_\rho - \frac{\hbar^2}{2m_{\text{r}}}\frac{\Delta_\Omega}{\rho^2} - E\right)\langle\boldsymbol{\rho}|\Psi\rangle = 0. \tag{83}$$

This equation is satisfied by the contact state everywhere except at $\rho = 0$ where the $N$-body wave function is singular and also at the contact of two particles interacting resonantly in the $s$-wave in the 3D (resp. 1D) space where Eq. (18) [resp. Eq. (20)] holds.

## 5.2 Log-derivative condition

In the low energy limit, one expects an equivalence between the contact model and the reference model valid at the order of the effective range approximation. The contact hyperradial function satisfies thus a log-derivative condition of the following form:

$$\left.\frac{\rho\,\partial_\rho F(\rho,E)}{F(\rho,E)}\right|_{\rho=R} = \epsilon - s - \theta R^2 k^2. \tag{84}$$

The parameter $\epsilon$ plays the role of a detuning parameter for the contact model and the length $R$ is denoted in what follows as the effective radius.

The log-derivative condition in Eq. (84) is a simple way to impose the asymptotic behavior of the contact states in the vicinity of the singularity at $\rho = 0$. This condition generalizes the energy independent condition used in Ref. [26]. It is an alternative way to define a contact model which is more usually defined by using a contact condition. The contact condition equivalent to Eq.(84) will be given in Sec. 5.6.

## 5.3 Determination of the parameters of the contact model

The three parameters $\epsilon$, $R$ and $\theta$ in Eq. (84), are such that in the low energy limit, the spectrum of the contact model coincides with the one of the reference model. A straightforward choice is given by

$$R = R_0, \qquad \epsilon = \delta, \qquad \theta = v'_0. \tag{85}$$

More generally, the equivalence of the two models is obtained by identifying the generalized scattering length and respectively the range parameter of the two models. Thus, the parameters $\epsilon$, $R$ and $\theta$ satisfy the equations

$$\xi_s = \frac{\pi(\epsilon - 2s)R^{2s}}{\epsilon s 4^s \Gamma(s)^2}, \tag{86}$$

$$\alpha_s = \frac{4^s \Gamma(s+1)^2 R^{2-2s}}{\pi(s-1)(2s-\epsilon)^2}(1 + 2(s-1)\theta), \tag{87}$$

where $\xi_s$ and $\alpha_s$ are given in Eqs. (44,45) and $|\epsilon| \ll 1$. Consequently, an infinite number of choices are possible for the three parameters of the contact model.

It is worth pointing out that Eqs. (86,87) can be also derived by identifying at the first order in $q^2 R_0^2$, the equations satisfied by the energy of the shallow bound state in Eq. (22) obtained in the two models. This is a consequence of the universality of the short distance behavior of the hyperradial function in the separability region.

### 5.4 Contact model in different resonant regimes

#### 5.4.1 One-parameter resonant regime

In this regime defined in Sec. 4.1.1, the range parameter is neglected and the contact model is a one-parameter theory defined for instance by the generalized scattering length $\xi_s$ with the natural choice $R = R_0, \epsilon = \delta, \theta = 0$. Using Eq. (86), other choices are possible where the parameters $(\epsilon, R)$ satisfy

$$\frac{R^{-2s}\epsilon}{\epsilon - 2s} = \frac{R_0^{-2s}\delta}{\delta - 2s} \, . \tag{88}$$

As depicted in Sec. 4.1.1, this one-parameter regime occurs only when $s < 1$ for vanishingly small values of the detuning $\delta$ or if Eq. (60) is satisfied.

#### 5.4.2 Two-parameter resonant regime

When the range term is not negligible, the parameters $(\epsilon, R)$ are solutions of the system of coupled equations given by Eq. (88) and by:

$$(1 + 2(s-1)\theta)\frac{R^{2-2s}}{(2s-\epsilon)^2} = (1 + 2(s-1)v_0')\frac{R_0^{2-2s}}{(2s-\delta)^2} \, . \tag{89}$$

For $s \geq 1$, the factor $(1 + 2(s-1)v_0')$ in the right-hand side of Eq. (89) is always positive, whereas when $s < 1$, it can be either positive or negative. This has a consequence on the possible choices of the parameter $\theta$. From Eq. (89), one has the following inequalities:

$$s > 1 \implies \theta > \frac{-1}{2(s-1)} \, , \qquad \qquad \text{case (a)}$$

$$s < 1 \implies \begin{cases} \theta < \dfrac{1}{2(1-s)} \, , & \text{if} \quad v_0' < \dfrac{1}{2(1-s)} \, , & \text{case (b)} \\[4mm] \theta > \dfrac{1}{2(1-s)} \, , & \text{if} \quad v_0' > \dfrac{1}{2(1-s)} \, . & \text{case (c)} \end{cases} \tag{90}$$

The one-parameter regime is at the frontier between the case (b) and the case (c) of Eq. (90).

**Standard resonant regime –** The standard regime corresponds to the cases a) and b) of Eq. (90), i.e. when

$$1 + 2(s-1)v_0' > 0 \, . \tag{91}$$

In the 3D mapping, this regime encompasses two-body high partial wave resonances and also broad s-wave resonances. From this point of view, it can be qualified as a 'standard resonant regime'. An example of reference model where Eq. (91) is satisfied, is also given by the square well model of Sec. (7). In this regime, whatever the value of the index $s$, it is always possible to set $\theta = 0$. With this choice of the parameter $\theta$, one recovers the log-derivative condition for the contact model introduced in Ref. [26], i.e.:

$$\left.\frac{\rho \, \partial_\rho F(\rho, E)}{F(\rho, E)}\right|_{\rho=R} = \epsilon - s \, . \tag{92}$$

From Eqs. (88,89) one finds when $|\delta| \ll s$ (i.e. when $s$ is not too close to the Efimov threshold):

$$\epsilon \simeq |1 + 2(s-1)v_0'|^{\frac{s}{1-s}} \delta , \tag{93}$$

$$R \simeq \left|1 + 2(s-1)v_0'\right|^{\frac{1}{2-2s}} R_0 . \tag{94}$$

The use of the absolute values in the right-hand side of Eqs. (93,94) will permit one to use these identities in the paragraph 5.4.2.

From Eq. (94) with the choice $\theta = 0$, one finds that

$$R < R_0 . \tag{95}$$

This last inequality was obtained by using the modified scalar product in Ref. [26] and is reminiscent of the Wigner bound obtained for high partial waves in two-body scattering.

**Anomalous regime –** This regime corresponds to the case c) in Eq. (90) and thus occurs when $s < 1$ and

$$1 + 2(s-1)v_0' < 0 . \tag{96}$$

This means that there is a large energy dependence in the log-derivative condition of Eq. (15). The 3D mapping of Eq. (14) for $s = 1/2$, shows that this resonant regime is analogous to a two-body narrow $s$-wave resonance where the effective range is negative.

In this case, it is not possible to map the reference model to a contact model with an energy-independent log-derivative condition. However, one can make the choice $\theta = \theta^\star$ where

$$\theta^\star \equiv \frac{1}{1-s} , \tag{97}$$

so that the parameters $R$ and $\epsilon$ are given by the same equations than in the standard regime where $\theta = 0$ in Eq. (89), but with the change

$$(1 + 2(s-1)v_0') \; \rightarrow \; |1 + 2(s-1)v_0'| . \tag{98}$$

Then for $|\delta| \ll s < 1$, one recovers again Eqs. (93,94). Nevertheless, the inequality in Eq. (95) is not satisfied for $\theta = \theta^\star$.

### 5.4.3 Large range parameter limit

The regime where $v_0'$ is large can occur in the anomalous resonant regime when $(s < 1)$ or in the standard resonant regime when $(s > 1)$. For $s < 1$, in the limit of a large value of $v_0'$ as shown in Sec. 4.2, the transition from the one-parameter to the two-parameter resonant regime occurs for a value of the index $s$ of the order of $1/2$. In this situation, even for a small but finite value of the detuning, the effective range correction and thus the deviation from the one-parameter regime may be important. In this regime it is essential to go beyond the approximation of Eqs. (93,94) and for simplicity one can use directly the parameters in Eq. (85).

## 5.5 Bound and quasi-bound states

By construction the bound (or quasi-bound) states of the reference and contact models coincide. The analysis done previously is however more general than what was done in Ref. [26]. This last study is available only in the standard resonant regime where the effective radius $R$ and the detuning $\epsilon$ can be defined with the choice $\theta = 0$.

Despite this equivalence, the parameters of the reference and of the contact models in the standard resonant regime are generally not the same. One has thus to be aware that keeping fixed one parameter or another in a asymptotic law may change the interpretation of the results. For instance with the choice $\theta = 0$ in the standard resonant regime for $s \gg 1$, one has a quasi-bound state of energy and a width $\Gamma$ given by:[4]

$$E_r \simeq 2(s-1)\epsilon E_R \,, \qquad \frac{\Gamma}{E_r} \simeq \frac{2\pi(s-1)4^{1-s}}{\Gamma(s)^2}\left(\frac{E_r}{E_R}\right)^{s-1} \,, \tag{99}$$

where $E_R$ is the characteristic energy associated with the effective radius $R$:

$$E_R = \frac{\hbar^2}{2m_r R^2} \,. \tag{100}$$

Equation (99) was obtained in Ref. [26] and coincides with the scaling law of Ref. [22] (see Eq. (1) in this last reference). In the same limit one obtains from Eqs. (64,66):

$$E_r \simeq \frac{E_0 \delta}{v_0'} \,, \qquad \frac{\Gamma}{E_r} \simeq \frac{\pi 4^{1-s}}{v_0'\Gamma(s)^2}\left(\frac{E_r}{E_0}\right)^{s-1} \,. \tag{101}$$

Even if Eqs. (99) and (101) are equivalent, depending on which parameter is considered as fixed in this large index limit may lead to a wrong interpretation. In the limit where $s$ is large and for a fixed value of $v_0'$, the energy $E_R$ tends to $E_0$, $\epsilon$ tends to zero and $(s-1)\epsilon$ tends to $\delta/(2v_0')$. One then finds the same position of resonance $E_r$. However, this behavior is not explicit in Eq. (99) and for increasing values of $s$, one can think about fixed values of the parameters $\epsilon$ and $E_R$ in this last equation. The situation is even worth for the width $\Gamma$, in which case despite the equivalence of the two expressions, one cannot replace abruptly $E_R$ by $E_0$ in Eq. (99). To conclude this discussion, the physical meaning of Eq. (101) is clearly more transparent than the results of the contact model obtained with the choice $\theta = 0$, because $v_0'$ is directly related to the behavior of the reference states at small hyperradius. Moreover, in Eqs. (99,100) there is no information about the physical high energy scale $E_0$ which fixes the limit of validity of the modeling itself.

## 5.6 Contact condition

### 5.6.1 Construction of the condition

The contact condition for the hyperradial function is a way to impose a specific behavior for a vanishing hyperradius such that one recovers the behavior of the hyperradial reference wave function at small but finite hyper radius of the order of $R_0$. Due to the universality of the behavior of the contact states in the limit of small hyperradius, one can make a reasoning at negative energy with $F(\rho, E) = K_s(q\rho)$ when $R_{\text{sup}} = \infty$. The contact condition at the order of the effective range approximation, is then obtained by finding the linear operator that performs the mapping of the small hyperradius limit of the contact hyperradial function $F(\rho, E)$ to the expression

$$\frac{1}{\xi_s} - \alpha_s q^2 + R_0^{-2s}C(s, -q^2 R_0^2) - \frac{q^{2s}}{\sin \pi s} \,. \tag{102}$$

Then, the contact states which all satisfy Eq. (57), belong to the kernel of this operator. For this purpose, one uses the behavior of the Macdonald function $K_s(x)$ when $x \to 0$:

$$K_s(x) = \frac{\Gamma(s)}{2}\left(\frac{x}{2}\right)^{-s}\left[1 - \frac{x^2}{4(s-1)} + \dots \frac{\Gamma(s-k)}{\Gamma(s)k!}\left(\frac{-x^2}{4}\right)^k + \frac{\Gamma(-s)}{\Gamma(s)}\left(\frac{x}{2}\right)^{2s} + \dots\right]. \tag{103}$$

---

[4]A factor 2 is missing in the expression of $\Gamma/E_r$ of Eq.(16) in Ref. [26].

The mapping is done for each term of Eq. (57) as follows

$$\left(\frac{q\rho}{2}\right)^{-s} \longrightarrow \frac{1}{\xi_s}, \tag{104}$$

$$-\frac{1}{s-1}\left(\frac{q\rho}{2}\right)^{-s+2} \longrightarrow -\alpha_s q^2, \tag{105}$$

$$\frac{\Gamma(-s)}{\Gamma(s)}\left(\frac{q\rho}{2}\right)^{s} \longrightarrow -\frac{q^{2s}}{\sin \pi s}, \tag{106}$$

and if $s > 1 + \eta$, as discussed previously, it is necessary to introduce a counter-term with

$$\frac{(-1)^k \Gamma(s-k)}{\Gamma(s)k!}\left(\frac{q\rho}{2}\right)^{2k-s} \longrightarrow \frac{(-1)^k q^{2k} R^{2(k-s)}}{\pi(s-k)}, \tag{107}$$

where $k = \lceil s \rceil$ if $s > \lfloor s \rfloor + \eta$ and $k = \lfloor s \rfloor$ otherwise. For convenience one uses the operator already introduced in Ref. [26]

$$\lim_{\rho \to 0} ]\rho^{\beta}, F(\rho)[ , \tag{108}$$

which gives the coefficient of the term $\rho^{\beta}$ in the expansion of $F(\rho)$ as $\rho \to 0$. In this way, one obtains the following contact condition for $s < 1 + \eta$:

$$\lim_{\rho \to 0} ]\frac{\rho^{-s}}{\xi_s} + 4(s-1)\alpha_s \rho^{2-s} + \frac{s(4\rho)^s \Gamma(s)^2}{\pi}, F(\rho)[ = 0, \tag{109}$$

and for $s \geq 1 + \eta$:

$$\lim_{\rho \to 0} ]\frac{\rho^{-s}}{\xi_s} + 4(s-1)\alpha_s \rho^{2-s} + \frac{s(4\rho)^s \Gamma(s)^2}{\pi} + \frac{4^k \Gamma(s)k! R^{2(k-s)}}{\pi \Gamma(s-k+1)} \rho^{2k-s}, F(\rho)[ = 0, \tag{110}$$

where $k = \lceil s \rceil$ if $s > \lfloor s \rfloor + \eta$ and $k = \lfloor s \rfloor$ otherwise.

By construction, the contact condition in Eq. (109) (or Eq. (110)) is equivalent to the log-derivative condition (84) when the calculations are performed at the order of the effective range approximation.

Interestingly, the contact condition in Eq. (109) allows for a continuous description of the spectrum at the effective range order, as a function of the index $s \in [0, 1 + \eta[$. This permits one to take into account the large range parameter limit for all these values of the index. In this manner the transition from the one-parameter regime to the regime of large range parameter of Sec. 4.2 can be studied accurately beginning from an index in the vicinity of the Efimov threshold $s = 0$. In contrast, in Ref. [26], the effective range approximation was not used systematically and the range was neglected for $s < \eta$ to avoid a spurious divergence near $s = 0$.

### 5.6.2 One-parameter resonant regime

In the one-parameter resonant regime defined only for $s < 1$, the range term can be neglected in the contact condition of Eq. (109) which can be rewritten

$$\lim_{\rho \to 0} ](2s-\epsilon)(R\rho)^s - \epsilon(R\rho)^{-s}, F(\rho)[ = 0. \tag{111}$$

In Eq. (111) there is a freedom in the choice of the pair of parameters $(\epsilon, R)$. At the order of the effective range approximation, any choice of the parameters satisfying Eq. (86) gives

the same results than if one takes $\epsilon = \delta$ and $R = R_0$. The contact condition in Eq. (111) is equivalent to imposing the following behavior on the contact hyperradial function as $\rho \to 0$

$$F(\rho) = A \times \left[ \left(\frac{\rho}{R}\right)^s \epsilon + (2s - \epsilon) \left(\frac{\rho}{R}\right)^{-s} \right] + \dots, \tag{112}$$

where the scalar $A$ depends on the state that one considers. The contact condition in Eq. (112) was first defined in Ref. [19, 20]. More recently, this contact model has been generalized to describe losses in the two-component Fermi gas [45].

One recovers two well-known behaviors. First, when $s = 1/2$, using the 3D mapping in Eq. (14) with the usual scattering length $a = \xi_{\frac{1}{2}}$, one recovers the Bethe-Peierls contact condition:

$$f(\rho) \propto \left(\frac{1}{a} - \frac{1}{\rho}\right) \text{ as } \rho \to 0. \tag{113}$$

Second, in the limit $s \to 0$, one recovers the contact condition for a two-body $s$-wave resonance in a 2D space with:

$$F(\rho) \propto \ln\left(\frac{\rho}{a_{2D}}\right) \text{ as } \rho \to 0, \tag{114}$$

where $a_{2D} = R e^{-1/\epsilon}$. One can notice that Eq. (88) gives in the limit $s \to 0$:

$$R e^{-1/\epsilon} = R_0 e^{-1/\delta}, \tag{115}$$

showing the freedom in the choice of the parameters $(\epsilon, \delta)$ in the expression of $a_{2D}$.

An alternative way to impose this contact condition is to use the pseudo-potential for a two-dimensional resonant $s$-wave interaction [46].

### 5.6.3   Integer values of the index $s$

For integer values of $s = n$, the contact model of a $N$-body isolated resonance is formally equivalent to a contact model for the 2D two-body problem with a resonant interaction in the $n$-th partial wave. In this case, the series of $F(\rho)$ contains terms of the form $\rho^n \ln(\rho q c_n)$.[5] The case $s = 0$ has been already studied. For $s = 1$, it is not possible to neglect the range term. For this value of the index $s$, the behavior of the hyperradial function in the vicinity of $\rho = 0$ is:

$$F(\rho) = A \left[ \frac{2}{\rho} + q^2 \rho \ln(q\rho c_1) \right] + \dots, \tag{116}$$

with $c_1 = \frac{1}{2} e^{\gamma - 1/2}$. From Eq. (79), the contact condition for $s = 1$ can be written

$$\lim_{\rho \to 0} \left] \frac{\epsilon}{2\rho R} - \rho R, F(\rho) \right[ = 0, \tag{117}$$

where the action of the operator in Eq. (108) applied on the logarithmic singularity is:

$$\lim_{\rho \to 0} \left] \rho, \rho \ln(\alpha\rho) \right[ = \ln(\alpha R) - \theta + \frac{1}{2}. \tag{118}$$

---

[5]For a interger value of the index $s = n > 0$, one has in the limit $z \to 0$

$$K_n(z) = \left(\frac{z}{2}\right)^{-n} \sum_{k=0}^{n-1} \frac{\Gamma(n-k)}{2k!} \left(\frac{-z^2}{4}\right)^k - \left(\frac{-z}{2}\right)^n \frac{\ln(zc_n)}{n!} + \dots,$$

with $c_n = 1/2 \times \exp(\gamma - (1 + \dots 1/n)/2)$.

This last operator and the contact condition in Eq. (117) are invariant in a change of $(R, \theta)$ satisfying Eqs. (88,89), which give in the limit $s \to 1$:

$$\frac{\epsilon}{R^2} = \frac{\delta}{R_0^2}, \qquad Re^{-\theta} = R_0 e^{-v_0'}. \tag{119}$$

If one wants to take into account the real part of the logarithmic singularity consistently for higher integer values of the index $s = n$, it is necessary to include higher derivatives (i.e. $v^{(n)}(0)$, $v^{(n-1)}(0) \dots$) in the log-derivative of Eq. (15). At the order of the effective range approximation, one can neglect these contributions. Nevertheless, as shown previously in Sec. 4.1.2, the imaginary part of the logarithm is essential for the description of quasi-bound states. It is thus necessary in the zero-range approach to extract the imaginary part of the logarithm correctly. Finally from Eq. (59), one can deduce the contact condition

$$\lim_{\rho \to 0} \Big] \frac{\rho^{-n}}{\xi_s} + 4(n-1)\alpha_s \rho^{2-n} + \frac{4^n \rho^n n!(n-1)!}{\pi}, F(\rho) \Big[ = 0, \tag{120}$$

with the following prescription for the operator in Eq (108):[6]

$$\lim_{\rho \to 0} \big] \rho^n, \rho^n \ln(\alpha \rho) \big[ = \ln(\alpha R), \qquad n \geq 2. \tag{121}$$

## 5.7 Modified scalar product

### 5.7.1 General expression and properties

Two contact states defined by the domain in Eq. (84) are not mutually orthogonal. Moreover the singularity in $\rho^{-s}$ of any contact state when $\rho \to 0$, leads to an arbitrarily large occupation of the small hyperradius region when $s \geq 1$. This normalization catastrophe and the non orthogonality problem was solved in the standard resonant regime by introducing a modified scalar product in Ref. [26], in the same spirit of what was done in the two-body contact model for a high partial wave resonance in Refs. [30,47]. This method can be extended to the present formalism. The $\theta$ invariance of the results will be fruitfully used to show the equivalence with the usual scalar product when one considers the reference model.

One begins with the wronskian equality for two contact eigenstates with two distinct energies $E_1 \neq E_2$ when $R_{\text{sup}} = \infty$. For this purpose, one uses the 2D radial Schrödinger equation for $F(\rho, E_1)$ in Eq. (10) multiplied by $\rho F(\rho, E_2)^*$ and subtracted by the complex conjugate of its analog obtained by the substitution $E_1 \leftrightarrow E_2$. Integration of this last equation between $\rho = R$ and $\rho = \infty$ gives:

$$\int_R^\infty \rho F(\rho, E_1) F(\rho, E_2)^* d\rho = \frac{\hbar^2 R}{2m_r(E_2 - E_1)} \times W\left[F(\rho, E_1), F(\rho, E_2)^*, \rho = R\right]. \tag{122}$$

Then, using the log-derivative condition in Eq. (84), one finds

$$\int_R^\infty \rho F(\rho, E_1) F(\rho, E_2)^* d\rho + R^2 \theta F(R, E_1) F(R, E_2)^* = 0. \tag{123}$$

The modified scalar product which ensures the orthogonality of two eigenstates of the contact model and solves the normalization catastrophe is deduced directly from Eq. (123). For two

---

[6]A term $\frac{1}{2n}$ was added in the right-hand side of the analog of Eq (121) in Ref. [26], but this term can be neglected at the order of the effective range approximation.

contact states $(|\Psi_1\rangle, |\Psi_2\rangle)$ with the respective hyperradial functions $(F_1(\rho), F_2(\rho))$ it is defined by:

$$(\Psi_1|\Psi_2)_0 = \int_{\rho>R} d\mu \langle\Psi_1|\boldsymbol{\rho}\rangle\langle\boldsymbol{\rho}|\Psi_2\rangle + R^2\theta F_1(R)^* F_2(R). \tag{124}$$

This scalar product is independent of the energy of the eigenstates and can then be used for all states in the domain defined by Eq. (84).

In the standard resonant regime of Sec. 5.4.2, one can choose the parameters $(R, \epsilon)$ such that $\theta = 0$ and the modified scalar product has the very intuitive form obtained in Ref. [26] where the parameter $R$ plays the role of a cut-off introduced in the usual scalar product:

$$(\Psi_1|\Psi_2)_0 = \int_{\rho>R} d\mu \langle\Psi_1|\boldsymbol{\rho}\rangle\langle\boldsymbol{\rho}|\Psi_2\rangle. \tag{125}$$

It is also of interest to consider the one-parameter regime. It is shown in Ref. [20] (pages 65-66) from a contact condition equivalent to Eq. (111), that the contact model is self-adjoint with respect to the usual scalar product in this regime. Thus in this regime, in principle there is no need to use a modified scalar product. Nevertheless, there is no contradiction with the preceding results in this particular case. Indeed, there are two possible conditions for having the one-parameter regime when $s < 1$. The first one, corresponds to the limit of vanishing value of the detuning $\delta \simeq 0$. One finds in this limit a negligible contribution $[\equiv O((qR_0)^{2(1-s)})]$ in the norm, of the small hyperradius region. Then at the lowest order in energy the modified scalar product coincides with the usual one. The second possible condition is given by Eq. (60). In this case, the small hyperradius region contribution is even smaller than in the previous case $[\equiv O((qR_0)^2)]$.

### 5.7.2 Invariance with respect to a change of the parameters

The invariance of the modified scalar product in a change of the parameters $(\epsilon, R, \theta)$ satisfying Eqs. (86,87) is exact only at the resonance threshold. For a finite detuning and finite energy there is a negligible variation as shown in the following lines.

Using Eqs. (86,87), the generalized scattering length and the range parameter remain unchanged by a variation of the parameters $(d\epsilon, dR, d\theta)$, when

$$\frac{Rd\theta}{dR} = (1 + 2(s-1)\theta)\left(1 + \frac{\epsilon}{1-s}\right). \tag{126}$$

From the definition in Eq. (124) and the log-derivative condition in Eq. (84) one then finds

$$\frac{d(\Psi|\Psi)_0}{RdR} = |F(R,E)|^2 \left(\frac{\epsilon}{1-s} + 2\theta^2 q^2 R^2\right). \tag{127}$$

At $R_{\text{sup}} = \infty$, the bound state wave function is given by Eq. (22) for all positive values of the hyperradius and the norm of the bound state $(\Psi|\Psi)_0$ is obtained from Eq. (A.7). Then in the limit $qR \ll 1$, one has when $s < 1$:

$$\frac{d(\Psi|\Psi)_0}{(\Psi|\Psi)_0} = \frac{dR}{R}\left(\epsilon + 2(1-s)(\theta qR)^2\right) \times O\left((qR)^{2-2s}\right), \tag{128}$$

and when $s > 1$,

$$\frac{d(\Psi|\Psi)_0}{(\Psi|\Psi)_0} = \frac{dR}{R}\left(\epsilon + 2(1-s)(\theta qR)^2\right) \times O(1). \tag{129}$$

As expected one obtains from Eq. (128) and Eq. (129), the invariance of the modified scalar product in the limit of a small detuning and vanishing energy ($|\epsilon|, qR \ll 1$).

### 5.7.3 Self-adjoint extensions

As one of the basis of quantum mechanics, the self-adjoint character of an Hamiltonian is a key property for any realistic system and the notion of self-adjoint extension is essential in many situations [48,49]. In the presence of an inverse square potential, the singular character of the contact solutions at vanishing hyperradius makes this property not trivial [50] and one has to refer to the theory of self-adjoint extensions of operators. For this purpose one considers the solutions of the hyperradial equation

$$\left(\partial_\rho^2 + \frac{1}{\rho}\partial_\rho - \frac{s^2}{\rho^2} \pm i\tau\right)F_\pm(\rho) = 0\,, \tag{130}$$

where $\tau > 0$. The solutions are of the form

$$F_\pm(\rho) = \mathcal{A}_\pm K_s\left(e^{\mp i\frac{\pi}{4}}\rho\sqrt{\tau}\right) + \mathcal{B}_\pm I_s\left(e^{\mp i\frac{\pi}{4}}\rho\sqrt{\tau}\right)\,. \tag{131}$$

The acceptable solutions are localized and thus must tends to zero for arbitrarily large hyperradius. This implies that $\mathcal{B}_\pm = 0$ in Eq. (131). The number of linearly independent solutions for the eigenvalue $+i\tau$ gives the deficiency index $n_+ = 1$ and for $-i\tau$, the deficiency index $n_- = 1$. Thus one has $n_+ = n_- = 1$, and following the Weyl-von Neumann theorem [48], there is a one-parameter family of self-adjoint extensions for the contact Hamiltonian. This assertion may appear puzzling as the contact theory of $N$-body resonances is defined through the log-derivative condition in Eq. (84) with the three parameters $\epsilon, R$ and $\theta$. The answer to this apparent paradox relies on the fact that the modified scalar product is also parameterized by the effective radius $R$ and the parameter $\theta$. Hence, the way to understand this issue is that for a given metrics, defined by the parameters $R$ and $\theta$, there exists a family of log-derivative conditions in Eq. (84) parameterized by only one parameter: the detuning $\epsilon$.

Considering the eigenstate $|\Psi(E)\rangle$, the self-adjoint character of the Hamiltonian is proven by showing that the difference,

$$\Delta = (\Psi(E_1)|H_0\Psi(E_2))_0 - (H_0\Psi(E_1)|\Psi(E_2))_0\,, \tag{132}$$

is zero for arbitrary values of the energies $E_1, E_2$. One finds

$$\Delta = \frac{\hbar^2 R}{2m_r}W\left[F(\rho, E_1)^*, F(\rho, E_2), \rho = R\right] + \theta R^2 (E_2 - E_1)F(\rho, E_1)^* F(\rho, E_2)\,, \tag{133}$$

and $\Delta = 0$ thanks to Eq. (84).

### 5.7.4 Equivalence with the usual scalar product in the reference model

Let us consider the contact state $|\Psi\rangle$ associated with the reference state $|\Psi^{\text{ref}}\rangle$. The equivalence between the two scalar products is obtained by making the choice of the parameters in Eq. (85) which gives the norm

$$(\Psi|\Psi)_0 = \int_{R_0}^\infty \rho|F(\rho)|^2 d\rho + v_0'|R_0 F(R_0)|^2\,. \tag{134}$$

One then recognizes in the right-hand side of this last equation the contribution of Eq. (27) so that one obtains

$$(\Psi|\Psi)_0 = \langle\Psi^{\text{ref}}|\Psi^{\text{ref}}\rangle\,. \tag{135}$$

As shown in the subsection 5.7.2, this last identity stays valid in the low energy limit at the order of the effective range approximation where the parameters $(R, \theta)$ satisfy Eqs. (86,87). Moreover, two contact eigenstates of different energies being by construction orthogonal with respect to the modified scalar product, one has the equivalence

$$(\Psi_1|\Psi_2)_0 = \langle \Psi_1^{\mathrm{ref}}|\Psi_2^{\mathrm{ref}}\rangle \,, \tag{136}$$

where $(|\Psi_1^{\mathrm{ref}}\rangle, |\Psi_2^{\mathrm{ref}}\rangle)$ are the reference states associated with the contact states $(|\Psi_1\rangle, |\Psi_2\rangle)$. In this equivalence, one has neglected the contributions where the wave function is not separable when $\rho = R_0$ as in Eq. (27).

The norm of the contact state in Eq. (134) has been derived for $R_{\mathrm{sup}} = \infty$. When $R_{\mathrm{sup}}$ is finite, contributions in the modified scalar product for $\rho > R_{\mathrm{sup}}$ coincide with those in the usual scalar product so that, the equivalence between the two scalar products is still valid when $R_{\mathrm{sup}} < \infty$.

## 6 Box model

To model in a simple way the effect of a finite value of $R_{\mathrm{sup}}$ in absence of two-body shallow bound states (typically there is a 3D $s$-wave resonance for part of the interacting pair of particles and $R_{\mathrm{sup}} = |a_{3\mathrm{D}}|$ with a finite but large and negative two-body scattering length $a_{3\mathrm{D}}$), one can consider the picture of one particle in a $d$-dimensional box of hyperradius $R_{\mathrm{sup}}$ with the condition $F(\rho = R_{\mathrm{sup}}, E) = 0$. This condition models the fact that for increasing values of $\rho$, starting from an hyperradius of the order $R_{\mathrm{sup}}$, the $N$-body state is no longer separable and its component on the hyperspherical harmonic involved in the separability region is gradually depopulated.

The hyperradial function is then

$$F(\rho, E) = \mathcal{A}\left[ K_s(q\rho) - \frac{K_s(qR_{\mathrm{sup}})}{I_s(qR_{\mathrm{sup}})} I_s(q\rho) \right]. \tag{137}$$

Using the log-derivative condition in Eq. (15) and taking into account that the second term in $I_s(q\rho)$ in the right-hand side of Eq. (137) is small with respect to the term in $K_s(q\rho)$ when $\rho = R_0$, one finds:

$$\left. \frac{z\partial_z K_s(z)}{K_s(z)} \right|_{z=qR_0} = -s + \delta + v_0' q^2 R_0^2 - sX \,, \tag{138}$$

with the small parameter

$$X = \frac{-K_s(qR_{\mathrm{sup}})}{qR_0 K_s(qR_0)K_s'(qR_0)I_s(qR_{\mathrm{sup}})} \,. \tag{139}$$

A typical solution is given in Fig. (3) with $s = 1$, $\delta = -0.01$ and $v_0' = 0.5$. One obtains a result qualitatively similar to the behavior of one branch of an Efimov spectrum as a function of $-1/a_{3\mathrm{D}} > 0$ with an endpoint at the three body continuum for a finite and negative value of the 3D scattering length $a_{3\mathrm{D}}$.

## 7 Square well model

The contact model gives the external part of the shallow resonant states and not the interior part in the region behind the kinetic barrier where the actual interactions are sufficiently attractive to induce a bound or a quasi-bound state. This explains the universality of the results.

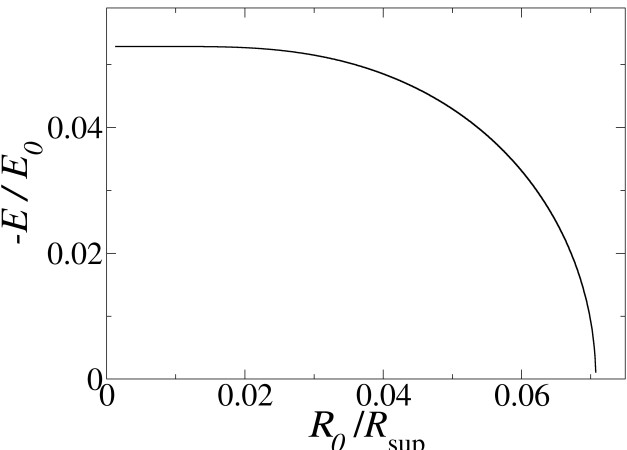

Figure 3: Example of spectrum obtained in the box model of Sec. 6, as a function of the inverse of the hyperradius $R_{\text{sup}}$ above which the bound state wave function is no longer separable.

It has been shown that the modified scalar product leads to the same normalization than the one obtained with actual finite range interactions (i.e. the reference model) in the limit of vanishing energy. This is why one obtains the generalization of the bounds found in Ref. [32] which have a general character. To have a simple illustration of these features, one considers in this section a simplified reference model of $N$-body resonance which is separable for all values of the hyperradius.

This model is defined by a square well potential at short distance without the effective centrifugal barrier in the range of the well:

$$
V(\rho) = \begin{cases} -\dfrac{\hbar^2 \kappa_0^2}{2m_{\text{r}}}, & \text{for} \quad \rho < R_0, \\ \dfrac{\hbar^2 s^2}{2m_{\text{r}} \rho^2}, & \text{otherwise}, \end{cases} \tag{140}
$$

and the hyperradial function for an eigenstate of energy $E$ is solution of

$$
\left[ -\frac{\hbar^2}{2m_{\text{r}}} \left( \partial_\rho^2 + \frac{1}{\rho} \partial_\rho \right) + V(\rho) - E \right] F(\rho, E) = 0. \tag{141}
$$

The bound state solution $E = -\frac{\hbar^2 q^2}{2m_{\text{r}}}$ of Eq. (141) is

$$
F(\rho, E) = \begin{cases} \mathcal{A}_{\text{in}} J_0(\kappa \rho), & \text{for} \quad \rho < R_0, \\ \mathcal{A}_{\text{out}} K_s(q\rho), & \text{otherwise}, \end{cases} \tag{142}
$$

where $\kappa = \sqrt{\kappa_0^2 - q^2}$. The continuity of the log-derivative at $\rho = R_0$ gives with the notation of Eq. (15):

$$
\left. \frac{z \partial_z J_0(z)}{J_0(z)} \right|_{z=\kappa R_0} = -\upsilon(-q^2 R_0^2). \tag{143}
$$

At the $N$-body resonance the value of $\kappa_0 = \kappa_0^{\text{res}}$ is obtained from Eq. (143) for $q = 0$ and $\upsilon_0 = s$ giving the equation

$$
z_0 J_1(z_0) = -s J_0(z_0), \tag{144}
$$

with $z_0 = \kappa_0^{\text{res}} R_0$ and from Eq. (144), one obtains

$$v_0' = \frac{1}{2} + \frac{s^2}{2z_0^2}. \tag{145}$$

Using the fact that the solutions of Eq. (144) are such that $\kappa_0^{\text{res}} R_0 > 2.4048\ldots$, one can verify that Eq. (91) is always satisfied and the resonance is in the standard regime for all possible values of $s$. By making the choice $\theta = 0$, the effective radius at resonance is given by:

$$R = \left(s - \frac{s^2}{z_0^2} + \frac{s^3}{z_0^2}\right)^{\frac{1}{2-2s}} R_0. \tag{146}$$

In Fig. (4), the solid black line is an example of a normalized hyperradial bound state solution of Eq. (141) computed at the detuning $\epsilon = -0.01$ and for the index $s = 1.3$. The contact solution is displayed in red in the region $\rho > R$ and in green for the small hyperradius part $\rho < R$. The contact state which is not square integrable has been normalized by using the modified scalar product. The two functions almost coincide for $\rho > R_0$. A similar analysis can be done if one includes the effective centrifugal barrier for $\rho < R_0$ with the potential

$$V(\rho) = \begin{cases} \dfrac{\hbar^2}{2m_{\text{r}}} \left(\dfrac{s^2}{\rho^2} - \kappa_0^2\right), & \text{for} \quad \rho < R_0, \\[3mm] \dfrac{\hbar^2 s^2}{2m_{\text{r}}\rho^2}, & \text{otherwise.} \end{cases} \tag{147}$$

Then, $F(\rho, E) = \mathcal{A}_{\text{in}} J_s(\kappa\rho)$ for $\rho < R_0$. One finds $v_0' = \frac{1}{2}$ and $R = (s)^{\frac{1}{2-2s}} R_0$, and the system is again in the standard resonant regime for all values of $s$.

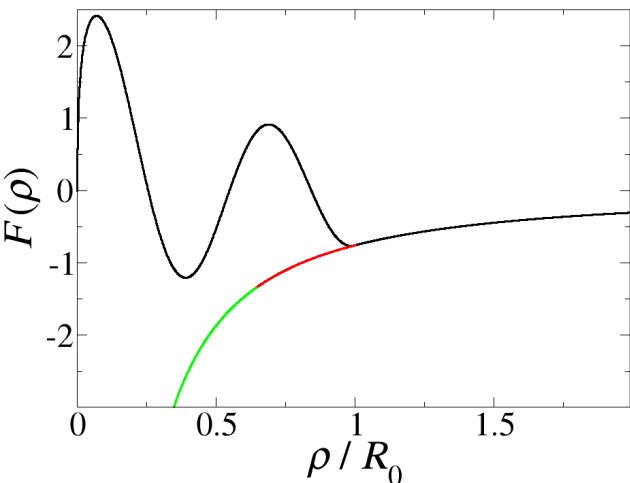

Figure 4: Example of hyperradial function for a shallow bound state in the vicinity of a resonance in the standard regime with $\epsilon = -0.01$ and $s = 1.3$. Black solid line: normalized reference function, solution of Eq. (141); red solid line: contact function for $\rho > R$ normalized by using the modified scalar product (it coincides with the black line for $\rho > R_0$); green solid line: contact function for $\rho < R$ (truncated at small hyperradius).

# 8  Conclusions

Various regimes of non efimovian $N$-body resonances near threshold have been explored, when a scale invariance can be identified. All the results are deduced from the properties of a potential with a repulsive tail identical to a kinetic barrier with an arbitrary strength. A contact model is then constructed to encompass all the possible situations. Three regimes emerge in the contact model: the single parameter regime which has been already introduced in Ref. [19, 20] when $s < 1$, the standard resonant regime which has been studied by means of different methods in Refs. [22–26], and the anomalous resonant regime. This last regime occurs for a large range parameter when $s < 1$ and can thus occur only in presence of two body $s$-wave scattering resonances. The identification of possible long-lived quasi-bound states in this last regime for small but finite detuning, when $\frac{1}{2} \lesssim s < 1$, is perhaps the most important physical result of this paper. A possible example for obtaining such values of the index $s$ is given by the problem of two fermions of mass $M$ interacting resonantly in the $s$-wave with an impurity of mass $m$ and a mass ratio in the interval $8.618 \cdots < \frac{M}{m} \lesssim 12.313 \ldots$ [51]. Similarly to two-body narrow (respectively broad) $s$-wave resonances [52], it is possible to show that a three-body resonance with a large (respectively small) range parameter will result from a Feshbach resonance mechanism in the small (respectively large) coupling limit between the three particles and a three-body molecular state in a closed-channel. This scenario which requires the tuning of both a two- and a three-body Feshbach resonance, paves the way of achieving the different regimes of resonances in the interval $0 \leq s \leq 2$.

The analysis of the three resonant regimes brings out the minimal number of $N$-body parameters needed to model $N$-body resonances. If a quasi-bound state exists at unitarity for a small and positive detuning then two three-body parameters are needed because two quantities must be parameterized: the energy and the width of the resonance. Likewise a two parameter law for the binding energy at negative detuning is deduced from the resonance energy by analytical continuation. A contrario, when a quasi-bound state cannot occur at positive detuning, then only one $N$-body parameter is needed in the small energy limit and one recovers the spectrum of Refs. [19, 20].

The threshold laws for bound and quasi-bound states have been obtained in this manuscript when the hyperangle-hyperradius separability and thus when the scale invariance is valid for arbitrary large scale. A box model has been introduced to have a qualitative approach when in a 3D space, a two-body scattering length $a_{3D}$ is large and negative (implying that the separability breaks down at an hyperradius of the order of, or larger than $|a_{3D}|$). In the examples of resonances considered in this manuscript (single resonant pair in nuclear halo states, $D\overline{D}\pi$ resonance and $^{3}$He droplets), the scattering length cannot be varied so that experimental results cannot be decisive to validate the universality issue (for instance considering a quasi-bound state, the two parameters laws can be always adjusted by using two data: resonance energy and width). In this respect, the possibility of varying the scattering length at fixed values of the three-body parameters using ultracold atoms is promising for studying the universality issue. The precise determination of the universal spectrum for given values of the three-body parameters as a function of the two-body scattering length is therefore essential. A possible candidate for experimental studies follows from the prediction of three-body resonances in presence of a two-body $p$-wave resonance [7] and by using the ($^{171}$Yb-$^{171}$Yb-Cs) system as suggested in Ref. [26].

At the heart of the present paper, the scattering problem of potentials with a tail in an inverse square law appears in various contexts [53]. In the repulsive case, the first thing that comes to mind is the centrifugal barrier in the two-body problem, which explains the Wigner's law at threshold in a scattering process and which has been generalized in Ref. [25] for the generic regime case defined in Sec. 2.1.1. It is also at the source of beautiful models initiated

by Ref. [54]. The fascinating attractive case leads also to predictions in very different area of physics: the Efimov effect [14] or the bound states of an electron with a polar molecule [55]. This paper deals with the repulsive case where bound or quasi-bound states exist for sufficiently attractive potentials at short distance. In the zero-range model picture, the scale invariance linked to the potential with a pure inverse square law is broken due to the contact condition, giving thus another example of quantum anomaly [50, 56, 57].

The contact model is a self-adjoint extension of the $N$-body Laplacian in association with a modified scalar product that solves the problem of non orthogonality and the normalization catastrophe. Beyond its application in the context of $N$-body resonances, this contact model is also interesting in itself as a new example of a way to handle a contact interaction leading to non square integrable localized states and thus this enriches the variety of contact models usually considered [58].

# Acknowledgments

It is a pleasure to acknowledge Yvan Castin, Pascal Naidon and Félix Werner for exciting and useful discussions on the subject.

# A  Useful relations used in this paper

This appendix gathers for convenience almost all the known properties of Bessel's and Gamma functions that are needed to derive the results in the main text.

To make the link between bound states and quasi-bound states in Sec. 3, one uses the analytical continuation

$$I_s(z) = i^{-s} J_s(iz), , \ K_s(z) = \frac{i\pi}{2} i^s H_s^{(1)}(iz),$$
(A.1)

together with the following relations between the Bessel's functions

$$J_s(z) = \frac{H_s^{(1)}(z) + H_s^{(2)}(z)}{2},$$
(A.2)

$$H_s^{(1)}(z) = \frac{i e^{-i\pi s} J_s(z) - i J_{-s}(z)}{\sin(\pi s)},$$
(A.3)

$$J_s(z)^* = J_s(z^*), \qquad H_s^{(1)}(z)^* = H_s^{(2)}(z).$$
(A.4)

To define the scattering phase shift in the inverse square potential, one uses the asymptotic behavior of the Hankel's function $H_s^{(1)}(z)$ for $z \to \infty$

$$H_s^{(1)}(z) \simeq \sqrt{\frac{2}{\pi z}} e^{i\left(z - \frac{\pi s}{2} - \frac{\pi}{4}\right)}.$$
(A.5)

The expressions of the generalized scattering length and of the range parameter in Eqs.(44,45) are deduced from

$$J_s(z) = \left(\frac{z}{2}\right)^s \sum_{k=0}^{\infty} \frac{\left(\frac{-z^2}{4}\right)^k}{k! \Gamma(s+k+1)}.$$
(A.6)

The small hyperradius behavior of the Macdonald function $K_s(z)$ which is used in the expression of the contact condition in Sec. 5.6 can be also obtained from this last series. One can

verifies also the identity

$$\int_z^{\infty} u K_s(u)^2 du = \frac{z}{2} W\left[z\partial_z K_s(z), K_s(z), z\right] = \frac{z}{2}\left[z K_{s+1}(z)^2 - z K_s(z)^2 - 2s K_{s+1}(z) K_s(z)\right], \quad \text{(A.7)}$$

which is used to obtain Eq. (30).

The two following properties of the Gamma function have been also used in the main text

$$\Gamma(z+1) = z\Gamma(z), \quad \text{(A.8)}$$

$$\Gamma(1-z)\Gamma(z) = \frac{\pi}{\sin(\pi z)}. \quad \text{(A.9)}$$

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
