# Peer review of "General features and contact modeling of $N$-body isolated resonances near threshold"

_SciPost Physics, doi:SciPost Phys. 17, 108 (2024)_

## Round 1 · Referee Report · Anonymous (Referee 2) · 2024-7-20

Report

The author has made the corrections suggested in my previous report, and has answered my question. I think the paper is ready for publication.

Recommendation

Publish (easily meets expectations and criteria for this Journal; among top 50%)

---

## Round 1 · Referee Report · Anonymous (Referee 1) · 2024-7-26

Report

The author has considerably improved the manuscript and addressed all
comments from the first report. He has also added a discussion of
6He as a possible two-neutron halo nucleus without a large s-wave neutron-core
scattering length. In this context, I remark that in the case of 6He there
is a very shallow p-wave resonance in the p_3/2 partial wave ("5He" channel)
which generates a large p-wave scattering volume of about -63 fm^3.
(See, e.g., the discussion in Bertulani et al., Nucl. Phys. A 712 (2002) 37
based on the phase shift analysis from Arndt et al., Nucl. Phys. A 209 (1973)
429.) This resonance is known to be important for the structure of 6He and
complicates the discussion of 6He in the framework of Refs. [8,9].
A brief comment on this complication in the paragraph on 6He would be
useful for the general reader not familiar with the details of 6He.
Apart from this minor issue, the manuscript is ready for publication in
SciPost Physics. I do not need to see the manuscript again.

Recommendation

Publish (easily meets expectations and criteria for this Journal; among top 50%)

---

## Round 1 · Author Response

Dear Editor,

In this new version of the manuscript, I have taken into account all the remarks and comments of both referees (see my previous reply to referee 1). I am very indebted to both of them for their relevant reports. They drew my attention on interesting issues (especially halo physics, regarding possible experimental perspectives and also the link with the work of Hammer and Lee). I am also very grateful to referee 2 for the improvements he made to the text.

My answer to the question 4 of referee 2 is as follows. The regime 1/2<s<1 can be reached for a system made of two identical fermions (mass M) interacting resonantly in the s wave with an impurity (mass m) for a mass ratio 8.61...<M/m<12.31... Then, similarly to a narrow s-wave two-body resonance, a large range parameter regime for a three-body resonance can be obtained if one considers a Feshbach resonance in the limit of a small coupling between the three particles and a three-body molecular state in a closed channel.

Best regards

---

## Round 1 · List of Changes

-All the text corrections of referee 2 have been implemented
-Minor typos / text changes have been implemented
-To simplify the reading, a change of notations has been implemented E_sep -> E_0 ; Rsep->R_0 ; k_0 -> k
-Eqs.(4,13) have been added (comments 2 and 3 of referee 2)
-Last paragraph of Sec. III-C has been added (comment 2 of referee 1)
-A technical precision on the parameter eta has been added in the paragraph before Eq. 58
-Last paragraph of Sec. IV-A-2 has been enlarged (comment 3 of referee 1)
-Last paragraph of Sec.IV-C-3 has been enlarged (comment 1 of referee 1)
-First paragraph of Sec. VII has been added
-The conclusion has been enlarged :
*paragraph 1 has been enlarged (comment 4 of referee 2 and comment 3 of referee 1) ; there was also an error on the interval for the mass ratio such that 1/2<s<1.
*paragraph 2 has been added
*paragraph 3 has been enlarged (answer to comment 3 of referee 1)
-Few references have been added [32,33,35,36,37,38,39,46,47,57,58]

---

## Round 2 · Author Response

Dear Editor,

I have implemented the remark suggested by one of the referees concerning the crucial role played by the p-wave resonant character of the interaction between one neutron and the alpha particle in explaining the binding of the 6He halo state. For readers interested in further details and explanations, I referred to the discussions of sections 3.7-3.10 in Ref. [4] and I have also added the reference C.A. Bertulani, H.-W. Hammer, U. van Kolck, Nucl. Phys. A 712, 37 (2002).

In the new pdf that I send to you, all the changes are written in red color.

I hope that in this form, the manuscript is ready for publication.

Thank you for your work.
Best regards,

Ludovic Pricoupenko

---

## Round 2 · List of Changes

-Suggestion of one of the referee concerning the crucial role played by the p-wave resonant character of the interaction between one neutron and the alpha particle in explaining the binding of the 6He halo state is implemented.

-Ref. C.A. Bertulani, H.-W. Hammer, U. van Kolck, Nucl. Phys. A 712, 37 (2002) has been added.

All the changes are in red color.

---

## Editorial Decision

published